# Hepatoenteric recycling is a new disposition mechanism for orally administered phenolic drugs and phytochemicals in rats

Yifan Tu[1], Lu Wang[1], Yi Rong[1], Vincent Tam[1], Taijun Yin[1], Song Gao[2], Rashim Singh[1], Ming Hu[1]*

[1]Department of Pharmacological and Pharmaceutical Sciences, College of Pharmacy, University of Houston, Houston, United States; [2]Department of Pharmaceutical Sciences, College of Pharmacy, Texas Southern University, Houston, United States

**Abstract** Many orally administered phenolic drugs undergo enterohepatic recycling (EHR), presumably mediated by the hepatic phase II enzymes. However, the disposition of extrahepatically generated phase II metabolites is unclear. This paper aims to determine the new roles of liver and intestine in the disposition of oral phenolics. Sixteen representative phenolics were tested using direct portal vein infusion and/or intestinal perfusion. The results showed that certain glucuronides were efficiently recycled by liver. OATP1B1/1B3/2B1 were the responsible uptake transporters. Hepatic uptake is the rate-limiting step in hepatic recycling. Our findings showed that the disposition of many oral phenolics is mediated by intestinal glucuronidation and hepatic recycling. A new disposition mechanism 'Hepatoenteric Recycling (HER)", where intestine is the metabolic organ and liver is the recycling organ, was revealed. Further investigations focusing on HER should help interpret how intestinal aliments or co-administered drugs that alter gut enzymes (e.g. UGTs) expression/activities will impact the disposition of phenolics.

*For correspondence:
mhu@uh.edu

## Introduction

Enterohepatic recirculation/recycling (i.e. EHR) refers to the recirculation/recycling of endogenous (produced within the body, like biliary acids and bilirubin) and exogenous (e.g. drugs or other substances) compounds. EHR involves the excretion of substances from the liver to the small intestine via bile, followed by absorption into enterocytes, which then enters the liver again via portal vein (*Lennartsson et al., 2012*). EHR is important in the disposition of both endogenous and exogenous compounds. For endogenous compounds (e.g. bile acids), the recirculation through EHR keeps them from being eliminated into the feces and reduces the requirement of daily biosynthesis. For exogenous compounds, EHR prolonged the in vivo exposure and increased their apparent half-life. In other words, EHR could greatly affect the efficacy of drugs and the toxicity of xenobiotics (or exogenous compounds).

EHR was first conceptualized in the 1950s for bile acid recirculation (*Norman and Sjovall, 1958*). The recirculation of bile acids requires only the re-absorption of the intact bile acids in the intestine without any change. In the 1960s, this concept was expanded into the disposition of drugs and other xenobiotics that undergo phase II metabolism via enzymes such as uridine 5'-diphospho-glucuronosyltransferase (UGTs) and sulfotransferases (SULTs) primarily expressed in the small intestine and liver. Currently, a variety of drugs are known to undergo EHR, including anticancer drugs (e.g. sorafenib, SN-38), anti-cholesterol drugs (e.g. ezetimibe), anti-osteoporosis drugs (e.g. raloxifene),

analgesics (e.g. morphine) *Gårdmark et al., 1993*, and others (e.g. mycophenolic acid) (*Trdan Lušin et al., 2012*; *Oswald et al., 2008*; *Ando and Hasegawa, 2005*; *Vasilyeva et al., 2015*; *Hasselström and Säwe, 1993*). In the case of exogenous compounds, the R in EHR was 'recycling' not 'recirculation,' as the enzymes of the microflora present in the intestine are required to hydrolyze the phase II metabolites and regenerate the aglycones (the parent form of the phase II metabolites) for reabsorption (*Williams et al., 1965*). The current concept of EHR does not define or specify the sources of phase II metabolites (i.e. where they are formed) that excreted in bile. Previously, it is postulated that the phase II metabolites (especially glucuronide metabolites) found in bile are mainly produced by the liver because bile acids are biosynthesized in the liver.

EHR of bile acids led most investigators, ourselves included, to believe that liver is the major organ contributing to the formation of conjugated metabolites. Faster in vitro microsomal conjugation rates observed in the liver as compared to intestinal microsomes for many compounds also supported this assumption (*Hu et al., 2014*; *Zhou et al., 2010*). However, we recently found that certain phenolic glucuronides formed in the intestine were efficiently taken up by hepatocytes and then rapidly excreted into the bile (*Zeng et al., 2016*). These phenomena suggest that glucuronides formed extrahepatically (such as in the intestine), available for hepatic uptake transporters (e.g. Organic Anion Transporting Polypeptides) and subsequently excreted by efflux transporters (e.g. Multidrug Resistance-associated Protein 2), could be recycled by the liver. This led us to hypothesize a new recycling phenomenon, 'Hepatoenteric Recycling (HER)'.

EHR differs from HER in the predominant organs for formation and transport of phase II metabolites. We believe that it is important to delineate HER from EHR since the disposition of phase II metabolites in different metabolic organs (such as liver and intestine) could greatly change with age, gender, and disease conditions (*Bolling et al., 2011*; *Bhatt et al., 2019*; *Liu et al., 2013*; *Zhou et al., 2013*; *Pimentel et al., 2019*; *Langmann et al., 2004*). These conditions often dictate the organ-level expression and activities of phase II enzymes and transporters (both efflux and uptake) that could significantly impact the EHR vs HER-mediated disposition of glucuronides. The changes in HER-mediated disposition can lead to changes in systemic and intestinal exposure of drugs and their glucuronides by altering the hepatic uptake and biliary excretion of glucuronides, which could significantly affect the drug efficacy and toxicity, both locally (intestinal lumen) and systemically.

However, EHR redefinition, which could explain the disposition of extrahepatically derived phase II metabolites, requires a systemic study to provide initial evidence that is based on the sources of biliary phenolic glucuronides. The current study provides answers to the three key questions: (1) how efficiently extrahepatically derived phenolic glucuronides are recycled by liver; (2) how these glucuronides are taken up by the liver cells; and (3) whether the aglycone or glucuronides entering the liver is better source of biliary excreted glucuronides. By answering these questions, we aim to elucidate the mechanism of HER, and the significance of distinguishing it from traditional EHR.

Through the analysis of our study results, a new disposition mechanism called 'Hepatoenteric Recycling (HER)' mainly applied to the disposition of drugs and their glucuronides, with significant intestinal metabolism, is proposed. The new mechanism distinguish from the enterohepatic recirculation/recycling (*Figure 1*), which is mainly applied to compounds which undergo recirculation in intact form or are metabolized by the liver. In HER, intestine is the organ for glucuronide formation and liver is the organ for its recycling (*Figure 1A*) as opposed to in EHR, where liver is the organ for glucuronide formation and intestine is the organ for its recycling (*Figure 1B*). The newly proposed HER concept delineates more clearly the disposition mechanism of phenolic drugs and phytochemicals such as flavonoids and polyphenols, which are significantly metabolized in the human intestine (up to ~70% or more) before entering the liver (*Patrick et al., 2002*; *Yang et al., 2012*; *Strassburg et al., 1998*; *Kokawa et al., 2013*; *Sun et al., 2013*; *Kemp et al., 2002*; *Kosoglou et al., 2005*; *Teeter and Meyerhoff, 2002*).

## Results

### Portal vein infusion

The portal vein infusion uses a direct method to assess the recirculation efficiency of a phenolic glucuronide by the liver using liver recycle efficiency% (LRE%, defined as steady-state biliary secretion

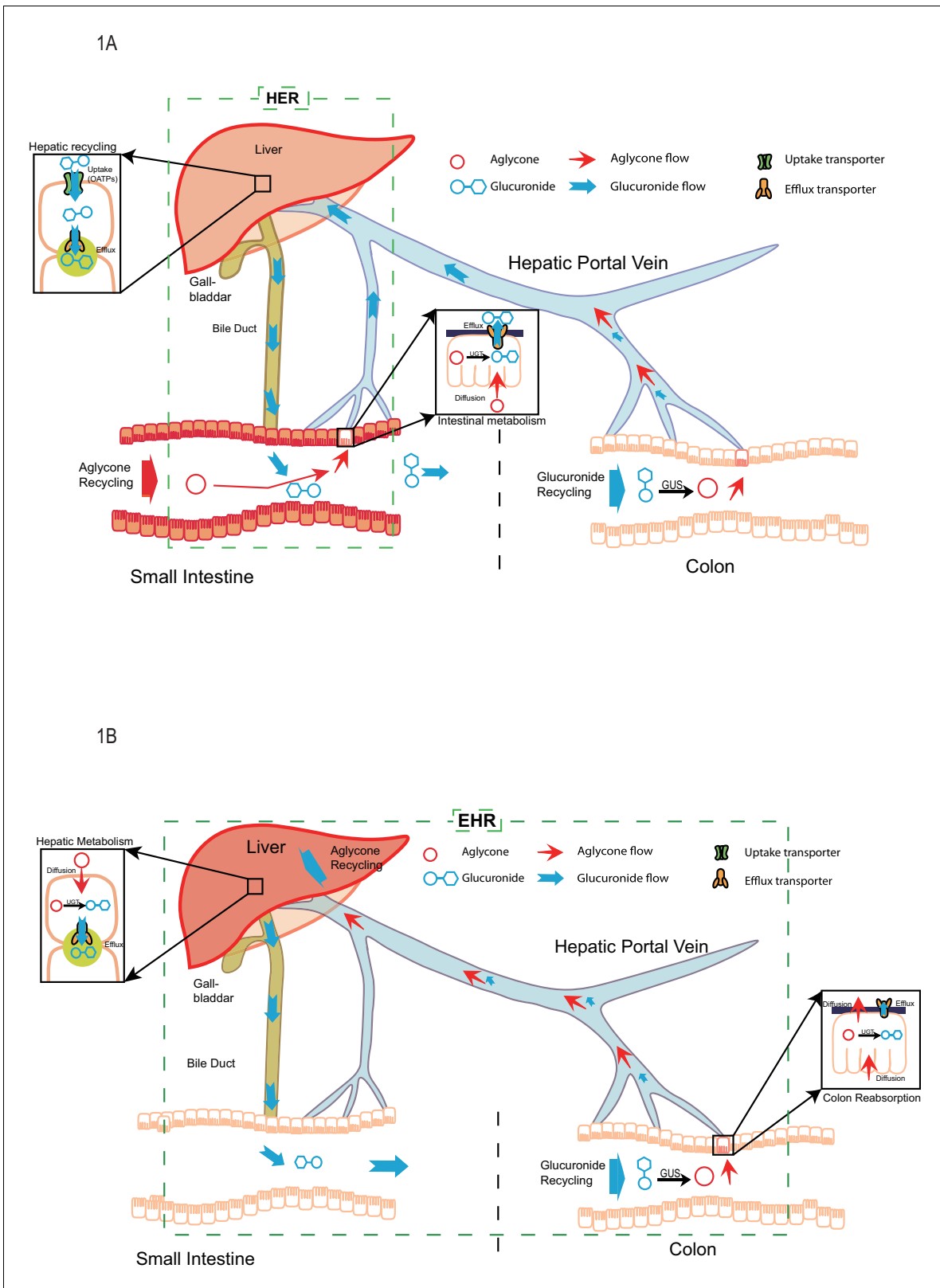

**Figure 1.** The pictorial representation of the newly proposed mechanism of hepatoenteric (HER) recycling and conventional mechanism of enterohepatic recycling/recirculation (EHR). (**A**). HER starts from intestinal glucuronidation of orally dosed aglycones. The glucuronides are taken up into the hepatocytes via portal vein using hepatic uptake transporter OATPs, and then excreted back into bile by apical hepatic efflux transporters, allowing them to return to the small intestine. For biliary glucuronides, the gut microflora β-glucuronides (GUS) will hydrolyze them back into the aglycone form,

*Figure 1 continued on next page*

*Figure 1 continued*

which are then reabsorbed in colon to complete the recycling. (B) EHR starts from the hepatic metabolism of an aglycone entering liver (from intestinal absorption or blood circulation) into its phase II metabolites (mostly glucuronides). The glucuronides formed in the liver are excreted into bile by the apical hepatic efflux transporters,returned to the small intestine and then moved to colon, where the gut microflora GUS hydrolyze them back into the aglycone form. The aglycone is then re-absorbed from colon and reached the liver again to complete the EHR.

rate divided by the hepatic infusion rate). Sixteen phenolic glucuronides, including four we studied previously (*Zeng et al., 2016*) and several of them with matching aglycones, were used to study this recirculating phenomenon.

## Effects of Won-7-G concentrations on LRE%

The biliary secretion and blood concentrations of Won-7-G reached steady state after 1 hr of infusion (*Figure 2A1 and A*) and remained steady until 2.5 hr. With increased Won-7-G concentrations, both biliary secretion amount (*Figure 2A1*) and blood concentrations (*Figure 2A2*) increased. LRE%, calculated by *Equation 1*, were significantly lower ($p<0.05$) at higher infused concentration (*Figure 2A3*). LRE% were also plotted against steady state blood concentrations (*Figure 2A4*). It showed that at lower blood concentrations (<100 µM) of wogonoside, its biliary excretion rates increased linearly. At higher blood concentrations (>=100 µM), biliary excretion rates showed a saturation trend, indicative of transporter-mediated excretion (*Figure 2A4*).

## Comparison between aglycones and their corresponding glucuronides

Lower LRE% were found for six aglycone compounds (including four flavonoids, and two drugs raloxifene and ezetimibe, *Figure 2B1–B7*) when comparing to their corresponding glucuronides. All seven glucuronides (raloxifene has two glucuronides) showed a significantly higher ($p<0.05$) LRE% than their corresponding aglycones (*Table 1*, *Figure 2B1–B7*). Among the aglycones, wogonin had highest LRE% (41%) when recycled in the form of Won-7-G. Other aglycones showed even lower LRE% (<10%), also in the form of their glucuronides. The generally lower LRE% of aglycones in comparison with their glucuronides were not expected since glucuronides are highly hydrophilic. To rule out the possibility that slow aglycone uptake was the reason for lower LRE%, an uptake comparison between Won and Won-7-G in three cell lines were also conducted (*Appendix 2—table 2* and *Appendix 2—table 3*). The results indicated that the uptake of Won was not influenced in the presence of specific transporter inhibitor, while Won-7-G was greatly inhibited, suggesting that aglycone uptake was mainly by passive diffusion, and often faster than their corresponding glucuronides.

## Effects of protein binding

It was reported that the protein binding of Won could be as high as 90% (*Talbi et al., 2014*); therefore, it is possible that lower LRE% values associated with aglycones was due to extensive binding to plasma protein (i.e. free aglycones were not readily available for liver uptake). To confirm this, we used higher concentration of aglycone (Won) that could saturate the plasma protein and more unbound aglycones could be available for hepatic uptake. However, the results indicated that LRE% at a higher concentration (100 µM) was significantly lower ($p<0.001$) than at a lower concentration (2 µM) (*Figure 2C3*), ruling out the impact of extensive protein binding on the lower LRE% of aglycones. Liver tissue concentrations of wogonin was at least five times higher than won-7-G concentrations (*Appendix 2—table 4*), which indicated that the slow metabolism in liver was the likely reason for the lower LRE%.

## Effect of phenolic and phenolic glucuronide structures on LRE%

We determined the LRE% of 16 phenolic-O-glucuronides, derived from dietary phenolics and phenolic drugs (*Appendix 2—table 5*). The results showed that LRE% was highly variable, ranging from 95% to 5%. In a study of nine 7-O-glucuronides of flavonoids, we were able to show that glucuronidation at 7-position alone is not sufficient to determine if a glucuronide will have high LRE%, suggesting that the aglycone structures are important determinant of its LRE%. This is rather interesting since there is only small structural differences between these flavonoid aglycones. On the other

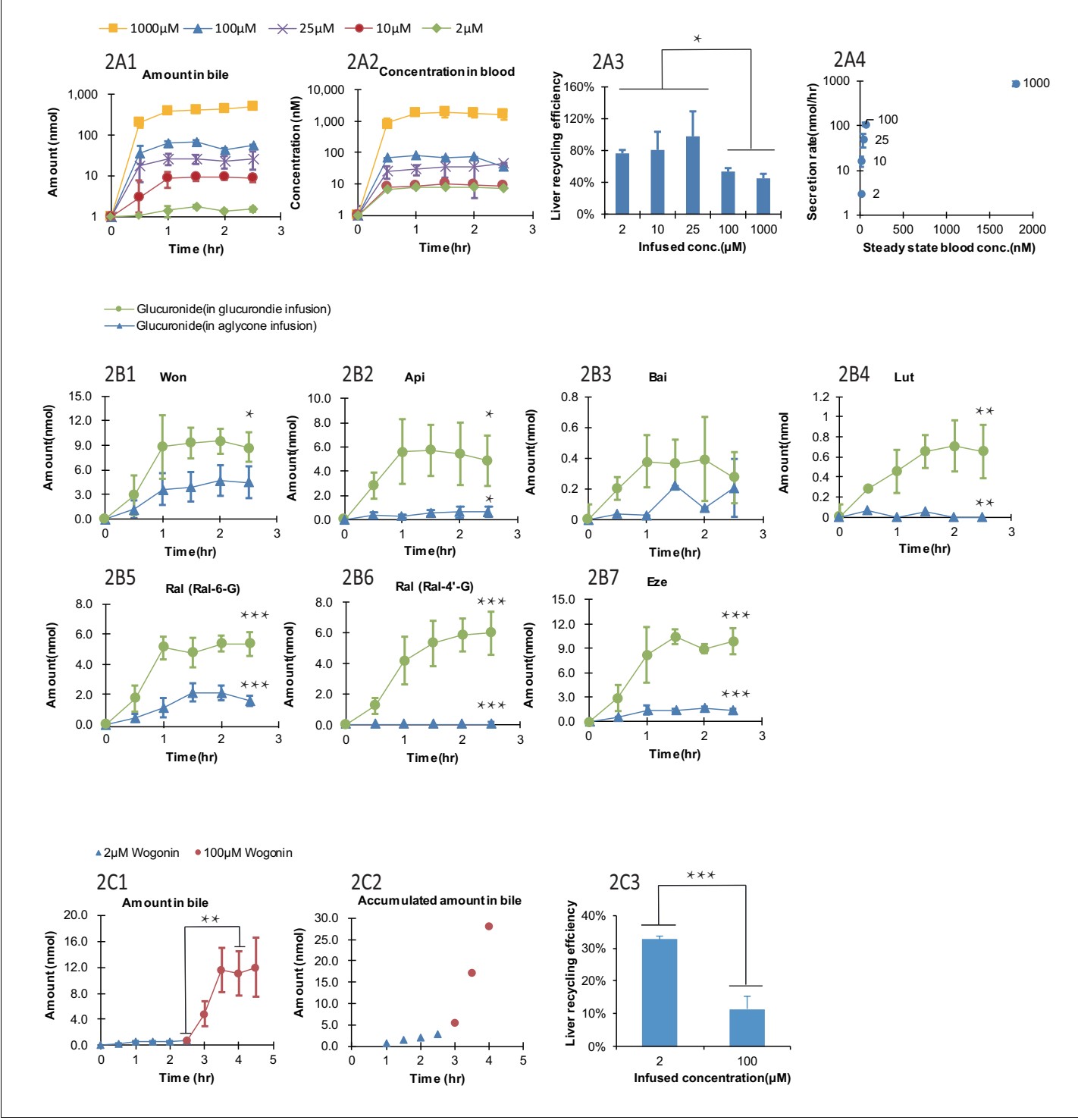

**Figure 2.** Effect of the glucuronide concentrations, aglycone structures, and aglycone concentrations on the biliary secretion, systemic exposure and liver recycling efficiency (LRE%) in a rat portal vein infusion model. Following portal vein infusion of Won-7-G at various concentrations (2–1000 µM), amounts of glucuronide excreted into bile (**A1**), concentrations of glucuronide in blood (**A2**) and LRE% (**A3**) were determined, and then the biliary secretion rates of glucuronides were plotted against their steady-state blood concentrations (**A4**). After the hepatic infusion of seven different aglycones [Won; Api; Bai; Lut; Ral; and Eze] and their corresponding glucuronides [Won-7-G; Api-7-G; Bai-7-G; Lut-3'-G; Ral-6-G; Ral-4'-G and Eze-4'-G] at 10 µM concentration, amount of the glucuronides secreted in bile during portal vein infusion of aglycones versus glucuronides were determined (**B1–B7**). Following the portal vein infusion of Won at 2 µM for the first 2.5 hr and at 100 µM for next 2 hr, the amount of Won-7-G secreted in bile (**C1**), the accumulated amount of Won-7-G secreted in bile (**C2**), and LRE% (**C3**) at low (2 µM) and high (100 µM) concentrations of Won were compared to study

*Figure 2 continued on next page*

*Figure 2 continued*

the effect of protein binding of Won on its liver uptake. The liver concentrations of Won and Won-7-G could be found in *Appendix 2—table 2*. 4. Four male Wistar rats were used in each experimental group. Statistical significance was calculated using student *t* test ('*', '**', and '***' indicates p<0.05, p<0.01, and p<0.001, respectively).

The online version of this article includes the following figure supplement(s) for figure 2:

**Figure supplement 1.** The bile amount and accumulative bile amount of Won-7-G.

**Figure supplement 2.** The microsome incubation and hydrolysis of Lut and its glucuronides.

**Figure supplement 3.** The blood concentration-time curve of Won-7-G(A), Bai-7-G(B), Eze-4'-G(C) and Ace-G(D) after orally administration of their corresponding aglycones, n = **4**.

hand, glucuronides of drug molecules showed similarly large difference in their LRE%, and their structures are quite different.

## Effect of sex differences on glucuronide LRE%

We are interested in determining the effects of sex difference on the OATP liver expression levels, because the expression level of Oatp in rat liver was reported to be significantly higher in males than in females (*Hou et al., 2014*). Also, E2G is a classical OATP substrate that has a drastically higher blood concentrations in females than males. The results indicated that there was not sex related differences in the LRE% of Won-7-G and Lut-3'-G. (*Appendix 2—table 6*). Interestingly, the LRE% of Bai-7-G increased from ~4% (male rats) to ~10% (female rats) (p<0.05).

## Effect of cassette dosing on glucuronide LRE%

To rapidly evaluate LRE% of glucuronides, we determined if cassette dosing of several glucuronides impacted their excretion when they were infused together. In order to avoid any potential

**Table 1.** Comparison of bililary secretion rates and recycle ratios of glucuronides following hepatic glucuronide infusion, hepatic aglycone infusion and small intestinal aglycone perfusion.

The rate of hepatic infusion was 20 nmol/hr and the rate of intestinal perfusion was 24 nmol/hr.

| Dosing method | Hepatic glucuronide infusion | Hepatic aglycone infusion | Small intestine aglycone perfusion |
|---|---|---|---|
| Infused compounds | Eze-4'-G | Ezetimibe (Eze) | Ezetimibe (Eze) |
| | Won-7-G | Wongonin (Won) | Wongonin (Won) |
| | Ral-6-G | Raloxifene (Ral) | Raloxifene (Ral) |
| | Api-7-G | Apigenin (Api) | Apigenin (Api) |
| | Bai-7-G | Baicalein (Bai) | Baicalein (Bai) |
| Measured compound | Bile secretion rate (nmol/hr) | | |
| Eze-4'-G | 19.31 ± 1.85 | 2.93 ± 0.41**\*,† | 21.94 ± 5.29 |
| Won-7-G | 16.30 ± 4.28 | 8.46 ± 3.93**\*,† | 17.90 ± 11.96 |
| Ral-6-G | 10.53 ± 1.51 | 3.91 ± 0.82*** | 4.00 ± 0.68 |
| Api-7-G | 10.64 ± 4.49 | 1.22 ± 0.76**\*,† | 11.32 ± 4.08 |
| Bai-7-G | 0.69 ± 0.29 | 0.09 ± 0.07**\*,† | ND† |
| Measured compound | Recycle ratio (%) | | |
| Eze-4'-G | 96.54 ± 9.23 | 14.65 ± 2.05**\*,† | 91.42 ± 22.04 |
| Won-7-G | 81.5 ± 21.41 | 42.30 ± 20.00**\*,† | 74.58 ± 49.83 |
| Ral-6-G | 52.64 ± 7.54 | 19.55 ± 4.10**\*,† | 16.67 ± 2.83 |
| Api-7-G | 53.21 ± 22.44 | 6.10 ± 3.80**\*,† | 47.17 ± 17.00 |
| Bai-7-G | 3.43 ± 1.46 | 0.45 ± 0.35**\*,† | ND‡ |

*Significant difference between hepatic glucuronide infusion and hepatic aglycone infusion, p<0.01.

†Significant difference between hepatic aglycone infusion and small intestinal perfusion , p<0.01.

‡Not determined due to below quantification limit.

competitive inhibition among glucuronides, we choose to infuse them at low concentration (10 μM). We found that when given at a low concentration (10 μM), cassette dosing did not significantly impact LRE% (*Figure 2—figure supplement 1A to D*). The results indicated that the hepatic recirculating system is capable of handling multiple glucuronides simultaneously at 10 μM concentration. This is because compounds infused into the portal vein were immediately diluted by ~200 folds therefore not expected to interact with each other (*Davies and Morris, 1993*).

## Cellular uptake with OATP overexpressing cells

We chose to focus on hepatic OATP transporters, because OATPs mediated the uptake of hormone glucuronides such as E2G (*Lin et al., 2012*). In addition, our earlier study had used glucuronides of isoflavones, which are phytoestrogens (similar to estrogens). OATP 1B1/1B3/2B1 overexpressing cells are used because they are expressed on the basolateral side (sinusoidal) membrane of hepatocytes.

### Structural effects on glucuronide uptake in OATP 1B1/1B3/2B1 cells

We found that uptake by three OATP cells varied greatly ($p < .005$) between glucuronides (*Figure 3A*). In addition, for a particular phenolic glucuronide, their uptake was also significantly different ($p < .005$) between three OATP-expressing cells. Moreover, most glucuronides with high LRE% was taken up rapidly in these cells. However, Lut-3'-G were found to have low LRE% (*Figure 2B4*) even though it was rapidly taken up by the OATP1B1 cells. The discrepancy can be explained by the fact that Lut-3'-G could be further metabolized into di-glucuronides of Lut in vivo (*Kemp et al., 2002*). A liver microsome-mediated metabolism of Lut-3'-G confirmed the formation of Lut di-glucuronides by LC-MS (*Figure 2—figure supplement 2A and B*). Moreover, the peaks of Lut-3'-G and Lut-di-G both decreased upon incubation with β-glucuronidases (*Figure 2—figure supplement 2C*).

### Concentration effects on glucuronide uptake

We determined the $K_m$ and $V_{max}$ values of four phenolic glucuronides (*Figure 2B1–2B4* and *Table 1*), representative of glucuronides with high (Won-7-G and Eze-4'-G), medium (Api-7-G), or low (Lut-3'-G) LRE%. We found that these glucuronides have good affinity to OATP1B1, 1B3, or 2B1 with $K_m$ values in the several micro molar range (*Appendix 2—table 7*), well within their in vivo exposure levels reported in the literature (*Mo et al., 2019*; *Wang et al., 2019*).

### Effects of OATP inhibitors on glucuronide uptake

Rifampicin (25 μM) was shown to be potent inhibitor (>90% inhibition) of the uptake of OATP1B1 and OATP 1B3 substrate E2G, Won-7-G, Lut-3'-G (all at 1 μM) (*Figure 3C1–C3*), consistent with literature-reports that rifampicin shown strong pharmacokinetic interactions with atorvastatin and pro-vastatin (classical OATPs substrate) in healthy volunteers (*Kashani et al., 2009*; *Riveros et al., 2009*). In addition to rifampicin, telmisartan was also shown to be a potent inhibitor (>90%) of the uptake of OATP 2B1 substrate E1S and Lut-3'-G (*Figure 3C3*). The results indicated that the uptake of these glucuronides was inhibited by specific OATP inhibitors. It supported our hypothesis that the uptake of these glucuronides was via a transporter-mediated process.

### Cross-inhibition of uptake between glucuronides and aglycones

To establish a fast screening criterion that determines whether a phenolic compound was likely to undergo HER as an OATP substrate, we performed the uptake cross-inhibition experiments using several known substrates and non-substrates of OATP1B1. The results showed that if a compound was not an inhibitor of prototypical OATP substrate, it was not a substrate of that OATP (*Figure 3D1–D3*), but the reverse was not true. In contrast, substrates of OATP1B1 were able to significantly inhibited ($p < .005$) the uptake of glucuronides (Won-7-G, E2G, and Lut-3'-G) that are good substrates of a specific OATP. Interestingly, aglycones were better inhibitors than their corresponding glucuronides, even though the corresponding glucuronides had higher LRE%.

### Correlation of LRE% and cell uptake

A total of 66 combinations with different OATP weightings were applied in Fisher exact test (*Appendix 4—table 1*). OATP 1B1 is the most abundant OATP expressed in livers (*Appendix 2—table 1*;

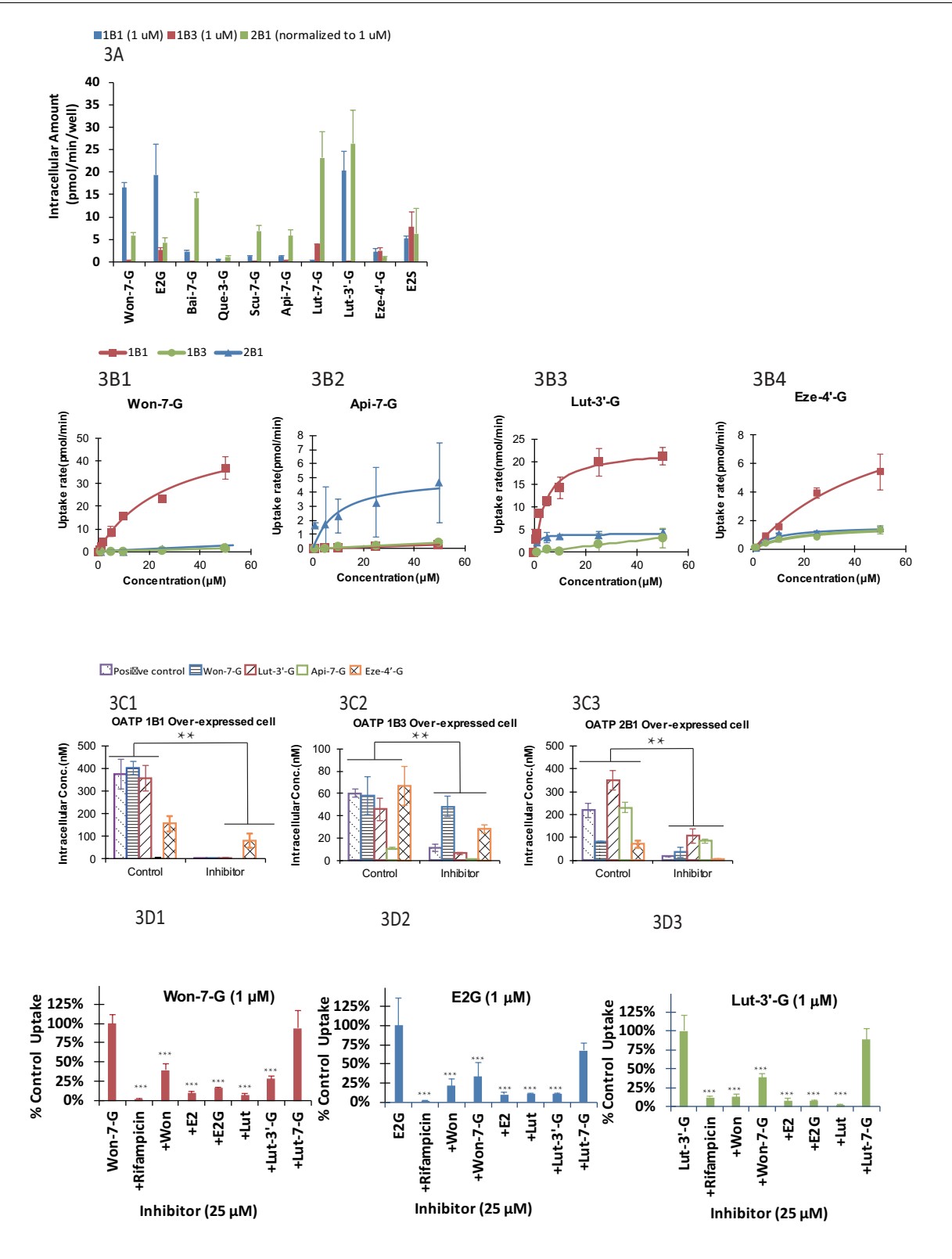

**Figure 3.** The OATP uptake kinetics, and the effect of glucuronide structures and uptake inhibitors on the hepatic uptake of glucuronides by OATP1B1/1B3/2B1 in over-expressed cell lines. Intracellular concentration of glucuronides obtained using 1 µM of 10 different glucuronides [Won-7-G; E2G; Bai-7-G; Que-3-G; Scu-7-G; Api-7-G; Lut-7-G; Lut-3'-G; Eze-4'-G; and E2S] (**A**) was determined (10 µM was used in OATP 2B1 but results were normalized to 1 µM). Uptake kinetics of Won-7-G (**B1**), Api-7-G (**B2**), Lut-3'-G (**B3**) and Eze-4'-G (**B4**) in the concentration range of 0.5–50 µM in OATP1B1/1B3/2B1 over-

*Figure 3 continued on next page*

*Figure 3 continued*

expected cell lines were determined. $K_m$ and $V_{max}$ values were calculated using Michaelis-Menten kinetics and summarized in *Appendix 2—table 7*. Effect of OATP inhibitors (50 µM rifampicin as OATP1B1/1B3 inhibitor and 50 µM erlotinib as OATP2B1 inhibitor) on the cellular uptake of five different glucuronides (E2G for OATP1B1/1B3 and E1S for OATP2B1 as positive controls) in OATP1B1 (C1), OATP1B3 (C2), and OATP2B1 (C3) over-expressed cell lines was studied at 10 µM substrate concentration (*Figure 3C1–C3*). Cross-over uptake in OATP1B1 cells using 1 µM of Won-7-G, E2G, Lut-3'-G as substrates was studied and intracellular concentrations of the glucuronides were determined in absence and presence of 25 µM of other aglycones and glucuronides as inhibitors. Each experiment was run in triplicate using substrates solutions in HBSS buffer (pH 7.4) at 37°C and the incubation lasted for 20 min. Statistical significance was calculated using student $t$ test in *Figure 3B1–B3* and one-way ANOVA in *Figure 3A and C1–C* ('*', '**', and '***' indicates $p < 0.05$, $p < 0.01$, and $p < 0.001$, respectively).

*Gårdmark et al., 1993*) and also considered the most important among all OATPs. OATP 1B3 is grouped with OATP1B1 in both localization and substrates, but the contribution of OATP 1B3 is considered to be always smaller than OATP 1B1. Therefore, combinations which assigned the highest weightings to OATP1B3 were considered not physiologically relevant and eliminated from the results. Of all the results from the Fisher exact test (*Appendix 4—table 2*), 22 combinations were eliminated based on this criterion. Of the remaining 44 combinations, 41 out of 44 results (93.2%) showed statistical significance. Taken together, these results demonstrated there were solid correlation between cellular uptake and LRE%. Thus, we showed that cellular uptake of glucuronides is the rate-limiting step in HER. In addition, this correlation (*Figure 4*) could be mathematically described by an Emax model (Emax86 ± 17%) with an EC50 value of 42 ± 33 nM.

## Impact of LRE% on the pharmacokinetic properties of phenolic compounds

The pharmacokinetic profiles of Won-7-G, Bai-7-G, Ace-G, and Eze-4'-G (*Figure 2—figure supplement 3A* to 3D) were fitted to a two-compartment model. $C_{max}$ values ranged from 41 nM (Eze-4'-G) to 572 nM (Ace-G), $T_{max}$ values ranged from ~2 hr (Ace-G) to ~8 hr (Eze-4'-G), and the half-lives ranged from 4.2 hr (Ace-G) to 52 hr (Won-7-G) (*Figure 2—figure supplement 1*). The elimination half-life was plotted against LRE% (*Figure 4*; *Figure 4—figure supplement 1*). Compound with a higher LRE% displayed a longer elimination half-life, which is expected since recycling will inevitably lead to a longer exposure.

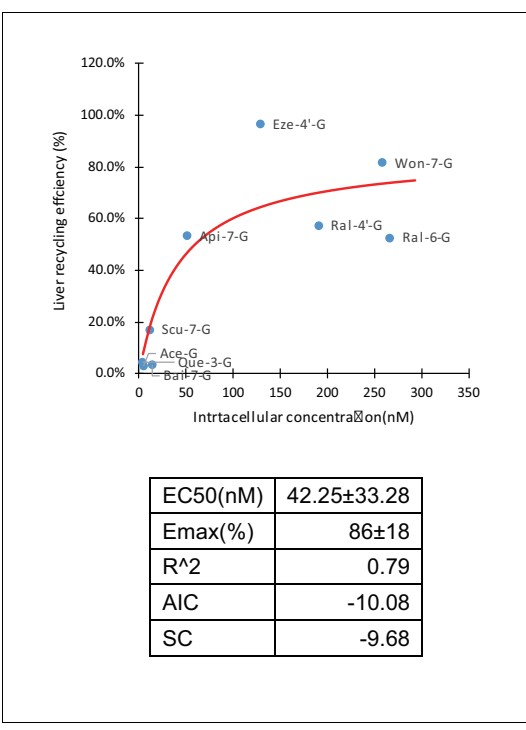

| | |
|---|---|
| EC50(nM) | 42.25±33.28 |
| Emax(%) | 86±18 |
| R^2 | 0.79 |
| AIC | -10.08 |
| SC | -9.68 |

**Figure 4.** Correlation of liver recycling efficiency (LRE%) and intracellular concentration of glucuronides. LRE% of 16 glucuronides calculated based on rat portal vein infusion experiment were plotted against the intracellular concentrations calculated as the sum of the individual measured intracellular concentrations in OATP1B1, 1B3, and 2B1 cells in the OATP uptake studies, weighted by their protein expression levels in human liver using $E_{max}$ model. The $E_{max}$ and $EC_{50}$ parameter values were estimated and summarized in the table below the graph.

The online version of this article includes the following figure supplement(s) for figure 4:

**Figure supplement 1.** The elimination half-life of Won-7-G, Bai-7-G, Eze-4'-G, and Ace-G were plotted against with their corresponding liver recycling efficiency (LRE), n = **4**.

**Figure supplement 2.** The relative blood stability of Won-7-G, Bai-7-G, and Api-7-G at three concentrations (2, 10,and 25 µM).

**Figure supplement 3.** The blood concentration-time curve of Won (**A**), Bai (**B**), Eze (**C**), and Ace (**D**) after orally administration of the aglycones, n = 4.

## Biliary secretion of five phenolic glucuronides in the intestine perfusion versus portal vein infusion model

The results indicated that the biliary excretion rates of five phenolic glucuronides were similar when five aglycones (Wog, Bai, Api, Eze, Ral) were given using the small intestine aglycone perfusion or when their corresponding glucuronides were given via direct portal vein infusion. Moreover, these rates were significantly higher than those observed with direct hepatic infusion of aglycones (*Table 1*), except for Ral. In intestinal perfusion study, the hepatic portal vein blood concentrations of glucuronides are also significantly higher (p<0.01) than tail vein blood (*Figure 5*). These results indicated that small intestine serves as a major metabolism organ for most of the perfused aglycones. Ral behaved differently in this study because Ral-6-G are formed both in liver and small intestine. The formation of Ral-6-G in liver is also fast. Taken together, the biliary excretion for most of phenolic glucuronides generated in small intestine was more efficient than direct portal vein administration of aglycones for four out of five phenolics.

## Effects of OATP inhibition on LRE% of phenolic glucuronides

In the presence of OATP transporter inhibitors (rifampicin, telmisartan, E2G and E1S at 1000 µM each with 1 hr pretreatment as well), the LRE% value of Won-7-G decreased from over ~80% to~40% and that of Lut-3'-G from ~7% to ~3% (*Figure 6A–D*). The results indicated that OATP transporters played the major role in the hepatic recycling of Won-7-G and Lut-3'-G, which is consistent with the results in the cell uptake study. According to the Fisher exact test, we claimed that hepatic uptake is the rate-limiting step in the recycling of these glucuronides. However, since we could not inhibit the LRE% by more than 80%, other uptake transporters could also participate in the recirculation of the glucuronides.

## Blood stability test of Won-7-G, Api-7-G, and Bai-7-G

After 2.5 hr of incubation, the relative stability of Won-7-G, Api-7-G, and Bai-7-G remained over 80% in 2 µM, 10 µM, and 25 µM (*Figure 4—figure supplement 2*). No significant changes were observed compared to the 0 hr time point. The results indicated that these glucuronide remained stable for the duration of the experiment time.

# Discussion

We have defined a new disposition process as 'Hepatoenteric Recycling (HER)' for phenolic drugs and phytochemicals (e.g. flavonoids) as well as their glucuronides (*Figure 1*). HER (*Figure 1A*) differs significantly from EHR (*Figure 1B*) associated with bile acids and certain drugs, in that intestine is the main metabolite forming organ and liver is the recycling organ. Previously, EHR of bile acids and various drugs (*Nielsen et al., 2009*) is associated with metabolites that are generated in liver and recycled in intestine. This new definition for the first-pass metabolism of phenolic drugs and phytochemicals pinpoints where the metabolites are formed and subsequently recycled. In other words, the roles of liver and intestine are reversed in the new HER vs the classical EHR. The word 'Recycling' refers to the fact that many extrahepatically generated phenolic glucuronides were able to achieve hepatic recycling in HER. The word 'Recycling' is also important in HER because biliary excreted glucuronides are not well absorbed in the intestine. They must be reactivated to aglycones, by the microflora β-glucuronidases, before reabsorption can occur (*Figure 1A*).

The new definition is needed because: (1) unlike EHR of bile acids, intestine, not liver, is the source of the recycled metabolites (i.e. phenolic glucuronides); (2) unlike EHR of bile acids, liver, not the intestine, is the recycling organ for glucuronides; and (3) unlike EHR of bile acids, hepatic (e.g. OATPs) but not intestinal uptake transporters (e.g. ASBT) appears to be the rate-limiting step in determining the LRE% in HER. Due to the presence of HER, hepatic transporters (e.g. OATPs) will often determine the plasma concentrations of glucuronides, and play a much more important role than the relevant hepatic enzymes (e.g. UGTs) in determining how much glucuronides are excreted into the bile and participate in recycling. For orally administered dietary phenolics such as wogonin and drugs such as ezetimibe with high LRE%, double peaks caused by HER were observed in their plasma profiles (*Figure 4—figure supplement 3A* to 3D) even though biliary metabolites are not generated by the liver. For phenolics with more moderate LRE%, the role of metabolite generating

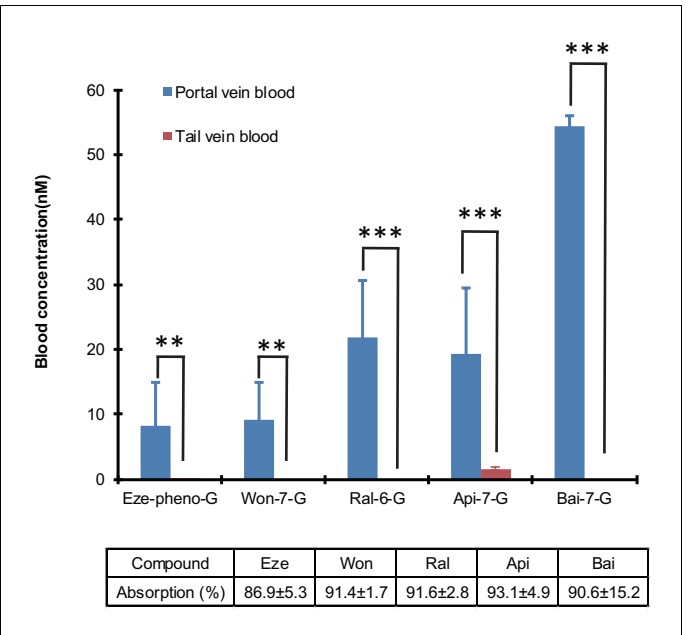

**Figure 5.** Differences in the portal vein and systemic concentrations of glucuronides after the intestinal perfusion of aglycone. five aglycones [Eze; Won; Ral; Api and Bai] were perfused in the rat small intestine (duodenum and jejunum, approximate length = 15 cm) individually at the rate of 24 nmol/hr for 2.5 hr in male Wistar rats (n = 4 per experimental group). Bile and intestinal perfusate samples were collected for every 30 min. Blood samples from tail vein and portal vein were collected at the end of the study and analyzed for the concentration of respective glucuronides. The absorption percentage of each aglycone were calculated and tabulated below the graph in the figure. The biliary secretion of the glucuronides was summarized in *Table 1*. Statistical significance was calculated using student t-test ('*', '**', and '***' indicates p<0.05, p<0.01, and p<0.001, respectively).

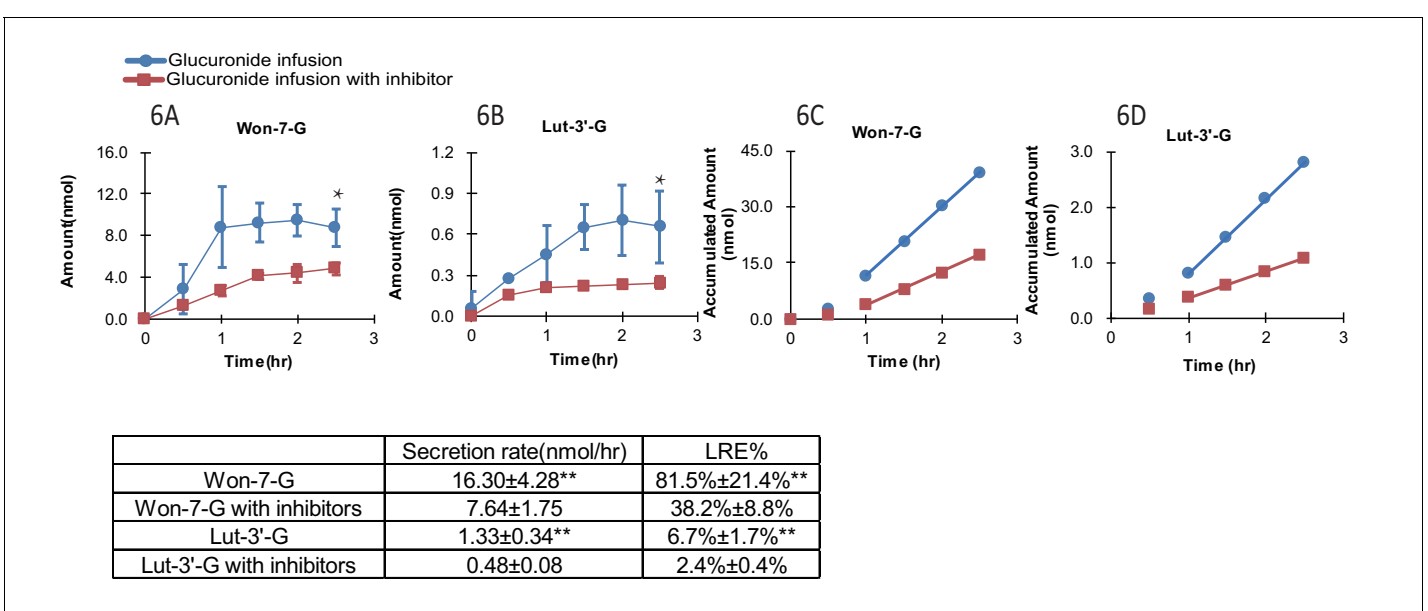

**Figure 6.** Effect of hepatic uptake inhibitors on the biliary excretion of glucuronides. A combination of four hepatic uptake inhibitors (rifampicin, telmisartan, estradiol-17β-glucuronide and estrone-3-sulfate) at 1 mM concentration were infused in rat portal vein for an hour as pretreatment, followed by co-infusion of inhibitors (at 1 mM) and glucuronide substrate [Won-7-G or Lut-3'-G] (at 10 μM) in the rat portal vein at the rate of 2 ml/hr for 2.5 hr in male Wistar rats (n = 4 per group). The bile amounts of Won-7-G (A), Lut-3'-G (B) and the accumulated bile amounts of Won-7-G (C), Lut-3'-G (D) with or without the uptake inhibitors were determined. Statistical significance was calculated by student t test ('**' indicates p<0.01).

organ or recycling organ is not mutually exclusive in HER vs EHR, in that biliary glucuronides may come from metabolites generated in the liver as well as the intestine, with variable contributions depending on the structure of aglycones.

How significant is this new disposition process called HER in determining how phenolics and their glucuronides are handled in vivo? We believe that HER is critically important in determining an orally administered phenolic's pharmacokinetic profile with extensive intestinal metabolism. A high LRE% means that a phenolic could have double peaks in their plasma profile and a longer terminal half-life. This could mean high plasma levels of aglycone if glucuronides are readily hydrolyzed back into aglycones (*Zhang et al., 2010*). In other words, for extrahepatically generated glucuronides, HER is critically important because almost every glucuronides we tested were more likely to be excreted into bile if they are given as glucuronides vs corresponding aglycones into the portal vein (*Figure 2B1–B7*). If HER is critical, how important is EHR in the recycling process? We believe that EHR is not that important here for compounds with high LRE%, since these compounds were nearly fully (>80%) metabolized after the aglycones were absorbed in small intestine (*Figure 5*). Therefore, we believe that HER is much more important than EHR for orally administered phenolics whose major metabolic pathway is rapid intestinal glucuronidation. The importance of HER can be further manifested in disease conditions. For example, in chemically (e.g. trinitrobenzene sulfonic acid) induced colitis model, small intestinal UGTs were significantly downregulated but hepatic UGTs were not (*Zhou et al., 2013*). These differences in UGTs regulation indicated that HER would likely to be impacted, whereas EHR might not, since changes in metabolism only occurs in the intestine. On the other hand, in hepatitis C, where hepatic CYPs, OATPs, OCTs (*Nakai et al., 2008*) and UGTs (*Smolders et al., 2016*) were all downregulated, EHR might be affected more than HER. However, for compounds that were metabolized both in the intestine and liver, the situation might be more convoluted.

We have proposed the new HER mechanism to delineate more clearly the significance and the need to describe the recycling of phenolics using a terminology different from EHR. How would HER impact the beneficial effects of dietary phenolics, therapeutic efficacy of phenolic drugs, and toxicities of phenolic toxins that are taken orally, produced by microbiota or formed as the result of phase II metabolism in the liver? The possible answers are made up of two parts, and subsequently combined to argue for the importance of HER in determining the disposition of orally administered phenolics with mainly extrahepatic glucuronidation.

Firstly, HER will shuttle more phenolic compounds and their metabolites, both beneficial and toxic to the colon. This is because there is no known mechanism that can rapidly take up glucuronides in the small intestine. For compounds with beneficial effects, including therapeutic efficacy, this new disposition mechanism can be useful in preventing or treating diseases in colon. The prevention or treatment effects can be achieved via a direct effect (e.g. suppress inflammation) or indirect effect (e.g. production of beneficial microbial metabolites that suppress inflammation). In contrary, some of the toxic phenolic metabolites can be further biotransformed by the microbial enzymes to more potent toxins (*Koppel et al., 2017*). Taken together, the presence of HER for an oral phenolic could substantially enhance its colonic impact and maintenance of a healthy colon microbiota (i.e. a desirable colon microbiome homeostasis) that is essential for our health. Therefore, the use of phenolics to maintain and improve the health of microbiota is an important future research goal. On the other hand, HER could promote the recycling of toxic phenolic metabolites, enhancing its exposure and toxicities, and this should be limited to improve human health.

Second, it may provide the theoretical foundation to understand a myriad phenomenon when there are drug-drug interactions involving glucuronidation. For example, if we only recognize the importance of EHR, if would be difficult to understand why inhibition of hepatic UGTs does not always increase aglycone concentration and decrease glucuronide concentration in the blood. If we recognize that a phenolic is mainly processed by HER, an inhibitor that only inhibit liver UGTs is not expected to affect plasma concentrations of orally administered aglycone and its glucuronides as long as intestine is the metabolite forming organ. In contrast, the emerging role of OATPs means that the plasma concentrations of both an aglycone and its glucuronide (if both are substrates of OATPs) or just glucuronide (if only glucuronide is substrate of OATPs), could increase by a drug that did not affect intestinal or liver UGT enzymes. Furthermore, one can propose a variety of drug-drug interaction schemes that can be explained by both HER and EHR. Therefore, the proposed HER mechanism should help us determine the dominating mechanism by which drug-drug interactions

occur to influence the plasma concentrations of a phenolic aglycone and its glucuronides that are formed in the intestine. This influence is more significant when the glucuronides are pharmacologically active (i.e. ezetimibe and dabigatran [*Jia and Zhao, 2011*; *Jiang and Li, 2010*]) or can inhibit CYP enzymes (i.e. gemfibrozil and clopidogrel [*Sivapathasekaran et al., 2010*; *Mutwil et al., 2010*]).

When combined together, the new HER definition could help predict the potential drug-drug interactions under pathological conditions. Since small intestinal UGTs play a significant role in the HER of the compounds, factors that alter the UGT expression in small intestine will significantly influence the recycling of the compounds and their in vivo exposure. Searching in Google Scholar by using key words combination 'UGT + intestine + disease', yielded a total of 14,500 hits. Among the top 50 results ranked by relevance, 15 of them reported different causes leading to change in intestinal UGTs including diseases (colitis, Gilbert's disease etc.), genetic polymorphism and drug-drug interactions. The results indicated that pathological conditions will greatly influence the intestinal UGTs expression level and thus change the in vivo exposure of a phenolic whose disposition is controlled by a HER process. Similar conclusion could be made on hepatic OATPs, since OATPs mediate the hepatic uptake step of glucuronides generated from intestinal metabolism. Using 'OATP +liver + disease' as key words combination, a total of 14,000 hits were found and 21 out of top 50 results ranked by relevance reported the hepatic OATPs expression changes due to disease like cholestasis, liver hepatitis, and carcinoma.

We have obtained several lines of evidence in support of new HER definition, especially for oral phenolics such as wogonin and ezetimibe. First, our data showed that when infused directly into the portal vein, the vast majority of their glucuronides reached bile, displaying a highly efficient recycling (LRE%) (*Appendix 2—table 5*). For those compounds, portal vein concentrations of glucuronides were much higher than the corresponding aglycones in the rat intestinal perfusion studies (*Figure 5*). Second, direct portal vein infusion of aglycone was substantially less effective in producing biliary excretion of glucuronides than their corresponding glucuronides (*Figure 2B1–B7*). Third, hepatic OATPs took up these glucuronides in a saturable process that can be inhibited by OATP inhibitors (*Figure 3B*, *Figure 3C1–C3*). Fourth, the structural differences between phenolic-glucuronides had a substantial impact on the LRE%. Taken together, these results showed clearly that intestine is the glucuronide forming organ, and liver is the glucuronide recycling organ for these two compounds. Hence, the term HER properly describes the determining disposition processes of these two compounds, a naturally occurring polyphenol flavonoid and a prescription drug, which share minimal structural similarity. The latter suggest that HER could be broadly applicable to other phenolic phytochemicals and drugs.

The new HER disposition process could explain the pharmacokinetic behaviors of many phenolic drugs, including ezetimibe, raloxifene and diclofenac. All three drugs were rapidly glucuronidated by intestinal microsomes and had a double peak PK profiles (*Watanabe et al., 1983*; *Herz and Bläsig, 1974*). For diclofenac, extensive intestinal glucuronidation by microsomes (*Bläsig et al., 1973*) with >95% glucuronides formation (*Herz and Bläsig, 1974*), are consistent with it undergoing through HER instead of EHR. In addition, the recycle properties of ezetimibe reported in human (*Christoffersen et al., 2015*) was consistent with the proposed HER hypothesis. As over 80% of ezetimibe was converted into glucuronides in intestine (*Christoffersen et al., 2015*) with a second peak in its PK profile attributable to hepatic recycling. Therefore, for compounds with extensive intestinal glucuronidation and hepatic recycling, HER will fully explain their pharmacokinetic behaviors.

Although many glucuronides with rapid uptake by OATPs are correlated with higher LRE% (*Figure 4*), two glucuronides (i.e. Lut-G and Ica-G) (*Wu et al., 2016*; *Wu et al., 2015*) did not. We found that these two glucuronides were further metabolized in liver, forming di-glucuronides of Lut and di-glucuronides of Ica, respectively, rendering the original glucuronides unavailable for biliary excretion. Whereas, we have obtained large amount of evidence in support of roles of OATP uptake transporters in enabling HER (*Figure 4*), we could not rule out the contribution of other transporters. This is because we could only achieve a maximum of 50% inhibition of Won-7-G and Lut-3'-G LRE% in portal vein infusion experiment (*Figure 6*). It is likely that transporters belong to the OAT3 subfamily contributed to their liver uptake, since several flavonoid glucuronides were shown to be substrates of OAT3 (*Rees et al., 2012*), which is also expressed in the liver (*Nema et al., 2011*).

Lastly, we have significant but not overwhelming structural diversity in our phenolic glucuronides, and as such, we could not predict which structure will have high LRE%. The empirical evidence

supports the hypothesis that good uptake of glucuronides by the OATPs will correlate with high LRE % , as long as the glucuronides taken up by the hepatocytes are not further metabolized. Because the direct portal vein infusion method is convenient to use with cassette dosing (*Figure 2—figure supplement 1*), HER potential can be readily estimated using OATP overexpressing cells and confirmed using the portal vein perfusion model. The latter should allow medicinal chemists to design new compounds with primarily intestinal glucuronidation that are tailored to treat intestinal, especially colonic diseases. In addition to the involvement of multiple uptake transporters, the role of efflux transporters may further complicate the disposition behaviors of phenolics and their glucuronides, but that is beyond the scope of this paper.

In conclusion, a new disposition terminology HER has been proposed to describe and delineate more clearly the disposition of dietary phenolics and phenolic drugs that are taken orally with intestine as their major metabolism (i.e. glucuronidation) organ and liver as their major recirculating organ. The new terminology more accurately captures the recycling of relevant phenolics, similar to the use of 'EHR' to capture the recycling of bile acids, where the liver is the metabolism organ and intestine as the recycling organ. This new HER more accurately captures the disposition of oral phenolics and can be used to better understand why this class of compounds may have larger than expected effects in the colon for human health and diseases. It may also help to delineate drug-drug interaction mechanisms involving intestinal UGT enzymes and hepatic transporters of glucuronides, a major challenge that we are still facing today because of all the complexity involved.

## Materials and methods

### Materials and reagents

Apigenin-7-O-glucuronide (Api-7-G) was purchased from HWI Analytik GmbH (Rheinzaberner, Rülzheim, German). Wogonin-7-glucuronide or wogonoside (Won-7-G), quercetin (Que-3-G), scutellarin (Scu-7-G), luteolin-3'-glucuronide (Lue-3'-G), luteolin-7-glucuronide (Lue-7-G), wogonin (Won), luteolin (Lue), icaritin (Ica), icaritin-3-glucuronide (Ica-3-G), and icaritin-7-glucuronide (Ica-7-G) were purchased from Meilunebio (Dalian, China). Estradiol-17β-glucuronide (E2G) and estrone-3-sulfate (E1S) were purchased from Steraloids Company (RI,USA). Raloxifene (Ral), raloxifene-4'-glucuronide (Ral-4'-G), raloxifene-6-glucuronide (Ral-6-G), ezetimibe (Eze), and ezetimibe-4'-phenoxy-glucuronide (Eze-4'-G) were purchased from Toronto Research Chemical (Toronto, Canada). Apigenin (Api), baicalein (Bai) and baicalein-7-glucuronide (Bai-7-G) was purchased from Indofine (NJ, USA). Dimethyl sulfoxide (DMSO) was purchased from Sigma-Aldrich (MO,USA). Acetonitrile (ACN) and methanol (MeOH) were purchased from Omni Solv (CA, US). ORA-Plus suspending vehicle was purchased from Perrigo (MI, USA). Hanks balanced salt solution (HBSS) was purchased from Sigma-Aldrich (MO, USA). All other chemicals were of reagent grade or better, which were purchased from reputable commercial suppliers.

### Animals

Experiments were performed on adult male and female Wistar rats weighing 280–330 g (male) or 220–270 g (female) at the time of the experiment. The rats were fasted for approximately 16 hr before the experiment. All procedures were approved by the Institutional Animal Care and Use Committee at the University of Houston.

### OATP-overexpressing cells

OATP1B1 and OATP1B3 over-expressed HEK-293 cells, described in a publication (*Musshoff and Daldrup, 1993*), were kindly provided by Dr. Yue Wei from University of Oklahoma Health Science Center. OATP2B1 over-expressed HEK-293 cell, first described in a publication (*Gjerde et al., 1991*), was kindly provided by Dr. Per Artursson's lab from Uppsala University (Uppsala, Sweden). Cell line authentication was conducted by Core lab in MD Anderson. All cell lines have been compared with the STR profiles in CCLC database and no contamination detected.

## Methods

### Bio-synthesis of flavonoid glucuronides

Selected flavonoid glucuronides were bio-synthesized using Hela-UGT1A9-MRP3 cells modified from the Hela-UGT1A9 cells developed in our lab (*Dougherty et al., 1990*). The concentration of glucuronide was quantified by UPLC using a previously published method (*Pérez-Gandía et al., 2010*).

### Portal vein infusion

The procedure of portal vein infusion was the same as our publication (*Zeng et al., 2016*). Briefly, rats were anesthetized by i.p. injection of 50% urethane at the dose of 1.875 g/kg. The portal vein and bile duct was catheterized after the anesthesia. The phenolic compounds or their corresponding glucuronides were prepared in HBSS buffer (pH = 7.4) and infused from portal vein catheterization at the rate of 2 ml/hr. Bile samples were collected from bile duct catheterization and blood samples were collected by snipping the tail. The infusion lasted for 2.5 hr and samples were collected every 0.5 hr. The liver recycling efficiency (LRE %) was calculated to evaluate the recycling efficiency using the following equation:

$$\mathrm{LRE(Liver\,recycling\,efficiency)}\% = \frac{\mathrm{Steady-state\,biliary\,glucuronide\,excretion\,rate}}{\mathrm{Portal\,vein\,infusion\,rate}} \quad (1)$$

The steady-state biliary glucuronide excretion rate was calculated by the linear regression analysis of accumulated amount excreted via bile vs. time curve.

This LRE%, derived from portal vein infusion experiments, was used to evaluate the effects of concentrations, inhibitors, structures of glucuronides on the liver recycling. Additional studies were used to determine the effects on LRE%, of aglycones vs glucuronides, protein binding, cassette dosing, and sex (male vs female).

### Effect of infusion concentrations on the LRE% of glucuronide using Won-7-G

Won-7-G was selected as a model compound. Won-7-G was prepared at 2, 10, 25, 100, and 1000 µM and infused into hepatic portal vein.

### Comparison of LRE% between aglycones and their corresponding glucuronides

Won, Lue, Api, Bai, Eze, and Ral and their glucuronides were selected and prepared at 10 µM concentration for the infusion experiments.

### Effect of aglycone concentrations on the LRE% of their corresponding glucuronides

The infusion study was conducted as described previously with minor modifications. After 2.5 hr of 2 µM Won infusion, the infusion solution was switched to 100 µM Won and infused for another 2 hr.

### Effect of structures on the LRE% of glucuronides

Thirteen glucuronides: Won-7-G, Api-7-G, Bai-7-G, Lut-3'-G, Que-3-G, Scu-7-G, Ica-3'-G, Ica-7-G, Lue-7-G, Eze-4'-G, Ace-G, Ral-4'-G, and Ral-6-G, prepared at 10 µM concentration were used in the infusion experiment to determine the structure activity relation of glucuronides and their recycle ratio.

### Effect of cassette dosing on the LRE% of glucuronides

A mixture of four glucuronides (Won-7-G, Api-7-G, Lut-3'-G, and Bai-7-G) were prepared at 10 µM together and infused simultaneously.

### Effect of sex differences on the LRE% of glucuronides

Won-7-G, Bai-7-G, and Lut-3'-G were prepared at 10 µM concentration. Prepared working solutions were further infused with female rats.

## Cellular uptake

### Cellular uptake of phenolic glucuronides by OATP over-expressed cell lines

Three OATP over-expressed cell lines (HEK 293 OATP 1B1/1B3/2B1 over-expressed cell lines) were used in the uptake study because OATPs are the most important hepatic uptake transporters for xenobiotics (Fahrmayr et al., 2010). Briefly, cells were seeded into 24 well plate 3 days before the experiment. Selected phenolic glucuronides were prepared in HBSS buffer (pH = 7.4) at required concentrations as working solution. Before incubation, cell culture medium was removed and cell was washed with 400 μl 37℃ HBSS buffer (pH = 7.4) twice. Working solution was incubated with cells at 37℃ for 20 min, when the intracellular concentrations reached steady state. After incubation, cells were washed with 400 μl ice-cold HBSS buffer (pH = 7.4) twice and cell pellet was collected in 200 μl of HBSS buffer. The cell pellet was further sonicated for 30 min. 150 μl of suspensions was collected and 150 μl of acetonitrile (contain 0.2 μM rutin as internal standard) was added into the pellet suspension. The suspension was centrifuged at 15,000 rpm for 15 min and supernatant was collected for analysis to measure the intracellular concentration of glucuronides.

### Effect of specific inhibitors on the cell uptake

Fifty μM OATP inhibitors (rifampicin for OATP 1B1/1B3 and erlotinib or telmisartan for OATP 2B1) were prepared in working solutions. Working solutions without inhibitors were set as control group. Cells were incubated with their corresponding working solutions (containing different inhibitors) for 20 min.

### Effect of substrate concentration on the cell uptake

Selected phenolic glucuronides were prepared at 0.5, 1, 5, 10, 25, and 50 μM as working solutions. Cells were incubated with working solutions for 20 min. The $K_m$ and $V_{max}$ values were calculated by using the Michaelis-Menten equation.

### Cross inhibition on OATP1B1 cells using glucuronides and their corresponding aglycones

Several known substrates and non-substrates of OATP1B1 was applied as an inhibitor to inhibit the uptake of three selected substrates (Won-7-G, E2G, and Lut-3'-G). Substrate concentration was 1 μM while inhibitor concentration was 25 μM. Cells incubated with substrate without inhibitor were set as control group (100% uptake). Relative uptake percentage was calculated by comparing intracellular concentrations of all experiment groups with the control group.

## Effects of OATP inhibition on the hepatic recycling of glucuronides

A cocktail consists of 1 mM rifampicin (OATP 1B1/1B3 inhibitor), 1 mM telmisartan (OATP 2B1 inhibitor), 1 mM E2G (OATP 1B1/1B3 substrate), and 1 mM E1S (OATP 2B1 substrate) were prepare in HBSS buffer (pH = 7.4) and used in the experiment. The experiment procedure was the same as we described earlier with minor modifications. To achieve a better inhibition effect, rats were infused with the inhibitor cocktail for 1 hr before the beginning of glucuronides infusion. After 1 hr of treatment, selected phenolic glucuronides were infused with inhibitor cocktail for 2.5 hr.

## Small intestine perfusion

To investigate the impact of intestinal metabolism on biliary excretion of phase II metabolites, small intestine perfusion study was performed as our previous publication with minor modifications (Jia et al., 2004). Briefly, one segment of small intestine was perfused. The perfusion solution started from the beginning segment duodenum and went through 15 cm length of down-stream intestine after the inlet segment. 2 μM of flavonoid aglycones (Won, Api, Bai) and phenolic drugs (Eze, Ral) were prepared together in HBSS buffer (pH = 7.4) as perfusion solution. The inlet cannulate was insulated and kept warm by a 37℃ circulating water bath. The perfusion rate was set at 0.193 ml/min. The perfusion lasted for 2.5 hr. Bile samples and perfusate were collected. At the last time point, blood samples from tail vein and from hepatic portal vein were also collected and analyzed. The absorption (%) was calculated as below:

$$\text{Absorption}(\%) = \frac{\text{Perfused amount of aglycone} - \text{amount of aglycone in perfusate}}{\text{Perfused amount of aglycone}} \quad (2)$$

## Evaluation of LRE% on the pharmacokinetic profiles of selected phenolics

To investigate whether the HER potentials have an impact on pharmacokinetic profiles of phenolic compounds, a pharmacokinetic experiment was conducted (n = 4). Briefly, Won and Bai were prepared together in oral suspension with ORA-Plus at the concentration of 50 mg/ml. Drugs were given to animals at the dose of 30 mg/kg. Eze/Ace were prepared and dosed the same way as described above for the other two flavonoids. Blood samples were taken at 0, 0.5, 1, 2, 4, 6, 8, and 24 hr after dosing. Samples were analyzed with LC/MS method after sample processing as detailed below.

## LC/MS analysis of blood, bile, liver tissue, cell, and perfusate samples

Blood, bile, liver tissue, perfusate, and cell samples were analyzed by LC/MS after sample process. Detailed sample process methods could be found in Supplement: Bioanalysis Method.

## Blood stability test of Won-7-G, Api-7-G, and Bai-7-G

The stability of Won-7-G, Api-7-G, and Bai-7-G in blood matrix were evaluated. The stability test was conducted for 2.5 hr at three concentrations (2, 10, and 25 µM, n = 3).

### Correlation between LRE% and cell uptake

The LRE% obtained from rat infusion experiment were correlated with cell uptake results. Since cell studies were conducted using human OATPs over-expressed cell lines, the analysis was performed with Fisher exact test (*Blevins and McDonald, 1985*), the details of which is in Supplement: Fisher Exact Test of Correlation Analysis.

An $E_{max}$ model were used to describe the correlation since LRE% has a maximum of 100% with the LRE% and intracellular concentrations to describe the correlation. The relative expression level close to human liver OATP isoform expression level (*Appendix 2—table 1*) was applied in the calculation of intracellular concentrations. The elimination half-life of Won/Bai and Eze/Ace were calculated using WinNolin and plotted against their LRE% value.

## Acknowledgements

This work is supported by NIH GM 070377 and CA246209.

## Additional information

### Funding

| Funder | Grant reference number | Author |
|---|---|---|
| National Institutes of Health | GM070377 | Ming Hu |
| National Institutes of Health | CA246209 | Ming Hu |

The funders had no role in study design, data collection and interpretation, or the decision to submit the work for publication.

### Author contributions

Yifan Tu, Conceptualization, Data curation, Software, Formal analysis, Validation, Investigation, Visualization, Methodology, Writing - original draft, Project administration, Writing - review and editing; Lu Wang, Data curation, Formal analysis, Validation, Investigation, Visualization, Methodology; Yi Rong, Resources, Data curation, Investigation, Methodology; Vincent Tam, Data curation, Formal analysis, Methodology, Writing - review and editing; Taijun Yin, Resources, Supervision, Investigation, Methodology; Song Gao, Supervision, Validation, Investigation, Methodology, Writing - review and editing; Rashim Singh, Resources, Formal analysis, Writing - review and editing; Ming Hu,

Conceptualization, Formal analysis, Supervision, Funding acquisition, Methodology, Writing - original draft, Project administration, Writing - review and editing

### Author ORCIDs
Yifan Tu https://orcid.org/0000-0002-2728-6753
Lu Wang http://orcid.org/0000-0002-8643-6790
Ming Hu https://orcid.org/0000-0002-2744-6294

### Ethics
Animal experimentation: This study was performed in strict accordance with the recommendations by the Institutional Animal Care and Use Committee (IACUC) at the University of Houston. The animal protocol (TR201800017) was approved by the IACUC at the University of Houston. All procedures were approved by the IACUC at the University of Houston. All surgery was performed under sodium pentobarbital anesthesia, and every effort was made to minimize suffering.

### Decision letter and Author response
Decision letter https://doi.org/10.7554/eLife.58820.sa1
Author response https://doi.org/10.7554/eLife.58820.sa2

---

## Additional files
### Supplementary files
• Transparent reporting form

### Data availability
All datasets associated with the published data will be made freely and widely available in the most useful formats, and according to the relevant reporting standards. All data generated or analysed during this study are included in the manuscript and supporting files and has been uploaded into Dryad Digital Repository. The files included the original version of the figures and tables included in the manuscript, appendix and supplement documents.

The following dataset was generated:

| Author(s) | Year | Dataset title | Dataset URL | Database and Identifier |
|---|---|---|---|---|
| Tu Y, Wang L, Rong Y, Tam V, Yin T, Gao S, Singh R, Hu M | 2021 | Dataset for manuscript EHR | http://dx.doi.org/10.5061/dryad.3ffbg79hn | Dryad Digital Repository, 10.5061/dryad.3ffbg79hn |

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

# Appendix 1

**Appendix 1—key resources table**

| Reagent type (species) or resource | Designation | Source or reference | Identifiers | Additional information |
|---|---|---|---|---|
| Chemical compound, drug | Apigenin-7-O-glucuronide (Api-7-G) | HWI Analytik GmbH | Lot#: 0449059 | |
| Chemical compound, drug | Wogonoside (Won-7-G) | Meilun bio | MB6662 | |
| Chemical compound, drug | Quercetin (Que-3-G) | Sigma-Aldrich | 00310590 | |
| Chemical compound, drug | Scutellarin (Scu-7-G) | Meilunebio | MB7004-S | |
| Chemical compound, drug | Luteolin-3'-glucuronide (Lue-3'-G) | Chengdu Alfa Biotechnology Co., Ltd. | AF8025306 | |
| Chemical compound, drug | Luteolin-7-glucuronide (Lue-7-G) | Chengdu Alfa Biotechnology Co., Ltd. | Af7022398 | |
| Chemical compound, drug | Wogonin (Won) | Meilunebio | MB6663 | |
| Chemical compound, drug | Luteolin (Lue) | Meilunebio | MB6799 | |
| Chemical compound, drug | Icaritin (Ica) | Meilunebio | MB7035 | |
| Chemical compound, drug | Icaritin-3-glucuronide (Ica-3-G) | | | This compound was synthesized by our cooperative lab. |
| Chemical compound, drug | Icaritin-7-glucuronide (Ica-7-G) | | | This compound was synthesized by our cooperative lab. |
| Chemical compound, drug | Estradiol-17β-glucuronide (E2G) | Steraloids Company | E1073-000 | |
| Chemical compound, drug | Estrone-3-sulfate (E1S) | Steraloids Company | E2335-000 | |
| Chemical compound, drug | Raloxifene (Ral) | Toronto Research Chemical | **R099995** | |
| Chemical compound, drug | Raloxifene-4'-glucuronide (Ral-4'-G) | Toronto Research Chemical | **R100020** | |
| Chemical compound, drug | Raloxifene-6-glucuronide (Ral-6-G) | Toronto Research Chemical | **R100025** | |
| Chemical compound, drug | Ezetimibe (Eze) | Toronto Research Chemical | **E975000** | |
| Chemical compound, drug | Ezetimibe-4'-phenoxy-glucuronide (Eze-4'-G) | Toronto Research Chemical | **E975030** | |
| Chemical compound, drug | Apigenin (Api) | Indofine | A-002 | |
| Chemical compound, drug | Baicalein (Bai) | Indofine | B-101 | |
| Chemical compound, drug | Baicalein-7-glucuronide (Bai-7-G) | Indofine | 06–012 | |
| Chemical compound, drug | Dimethyl sulfoxide (DMSO) | Sigma-Aldrich | 276855 | |
| Chemical compound, drug | Hanks balanced salt solution (HBSS) | Sigma-Aldrich | H1387 | |

*Continued on next page*

*Appendix 1—key resources table continued*

| Reagent type (species) or resource | Designation | Source or reference | Identifiers | Additional information |
|---|---|---|---|---|
| Chemical compound, drug | Acetonitrile (ACN) | Omni Solv | AX0149 | |
| Chemical compound, drug | methanol (MeOH) | Omni Solv | MX0486 | |
| Chemical compound, drug | ORA-Plus suspending vehicle | Perrigo | 0574-0303-16 | |
| Cell line (*Homo-sapiens*) | HEK-293 OATP1B1/1B3 over-expressed cell line | Dr.Yue Wei's Lab | | The related publication could be searched by using PMID:29538325 |
| Cell line (*Homo-sapiens*) | HEK-293 OATP2B1 over-expressed cell line | Dr. Per Artursson's Lab | | The related publication could be searched by using PMID:24799396 |
| Software, algorithm | GraphPad Prism 6 | GraphPad Software Inc | RRID:SCR_002798 | |

# Reagents

Chemicals and reagents used in this manuscript were summarized below. Detailed supplier information could also be found in this section.

# Materials and reagents

Apigenin-7-O-glucuronide (Api-7-G) was purchased from HWI Analytik GmbH (Rheinzaberner, Rülzheim, German). Wogonin-7-glucuronide or wogonoside (Won-7-G), quercetin (Que-3-G), scutellarin (Scu-7-G), luteolin-3'-glucuronide (Lue-3'-G), luteolin-7-glucuronide (Lue-7-G), wogonin (Won), luteolin (Lue), icaritin (Ica), icaritin-3-glucuronide (Ica-3-G), and icaritin-7-glucuronide (Ica-7-G) were purchased from Meilunebio (Dalian, China). Estradiol-17β-glucuronide (E2G) and estrone-3-sulfate (E1S) were purchased from Steraloids Company (RI,USA). Raloxifene (Ral), raloxifene-4'-glucuronide (Ral-4'-G), raloxifene-6-glucuronide (Ral-6-G), ezetimibe (Eze), and ezetimibe-4'-phenoxy-glucuronide (Eze-4'-G) were purchased from Toronto Research Chemical (Toronto, Canada). Apigenin (Api), baicalein (Bai) and baicalein-7-glucuronide (Bai-7-G) were purchased from Indofine (NJ, USA). Dimethyl sulfoxide (DMSO) was purchased from Sigma-Aldrich (MO,USA). Acetonitrile (ACN) and methanol (MeOH) were purchased from Omni Solv (CA, US). ORA-Plus suspending vehicle was purchased from Perrigo (MI, USA). Hanks balanced salt solution (HBSS) was purchased from Sigma-Aldrich (MO, USA). All other chemicals were of reagent grade or better, which were purchased from reputable commercial suppliers.

## Appendix 2

### Methods

Detailed descriptions of experiment design and operation were summarized in this section. All the experiments including animal infusion/perfusion experiment, cell uptake experiment and PK experiments were summarized below.

### 1. Portal vein infusion

Portal vein infusion

The procedure of portal vein infusion was the same as those described previously (*Zeng et al., 2016*). Briefly, rats were anesthetized by i.p. injection of 50% urethane at the dose of 1.875 g/kg. The portal vein and bile duct were catheterized after the anesthesia. The phenolic compounds or their corresponding glucuronides were prepared in HBSS buffer (pH = 7.4) and infused from portal vein catheterization at the rate of 2 ml/hr. Bile samples were collected from bile duct catheterization and blood samples were collected by snipping the tail. The infusion lasted for 2.5 hr and samples were collected every 0.5 hr. The liver recycling efficiency (LRE %) was calculated to evaluate the recycling efficiency using the following equation:

$$\mathrm{LRE(Liver\,recycling\,efficiency)\%} = \frac{\mathrm{Steady-state\,biliary\,glucuronide\,excretion\,rate}}{\mathrm{Portal\,vein\,infusion\,rate}} \quad (A2\text{-}1)$$

The glucuronide secretion rate at the steady state was calculated by the linear regression of accumulated amount excreted in bile vs. time curve.

### Effect of infusion concentrations on the LRE% of glucuronide using Won-7-G

Won-7-G was selected as a model compound. Won-7-G was prepared at 2, 10, 25, 100, and 1000 µM and infused into hepatic portal vein.

### Comparison of LRE% between aglycones and their corresponding glucuronides

Won, Lue, Api, Bai, Eze, and Ral and their glucuronides were selected and prepared at 10 µM concentration for the infusion experiments.

### Effect of aglycone concentrations on the LRE% of their corresponding glucuronides

The infusion study was conducted as described previously with minor modifications. After 2.5 hr of 2 µM Won infusion, the infusion solution was switched to 100 µM Won and infused for another 2 hr.

### Effect of structures on the LRER% of glucuronides

Thirteen glucuronides: Won-7-G, Api-7-G, Bai-7-G, Lut-3'-G, Que-3-G, Scu-7-G, Ica-3'-G, Ica-7-G, Lue-7-G, Eze-4'-G, Ace-G, Ral-4'-G, and Ral-6-G, prepared at 10 µM concentration were used in the infusion experiment to determine the structure activity relation of glucuronides and their liver recycling efficiency.

### Effect of cassette dosing on the LRER% of glucuronides

A mixture of 4 glucuronides (Won-7-G, Api-7-G, Lut-3'-G, and Bai-7-G) were prepared at 10 µM together and infused simultaneously.

### Effect of gender differences on the LRE% of glucuronides

Won-7-G, Bai-7-G, and Lut-3'-G were prepared at 10 µM concentration. Prepared working solutions were further infused with female rats.

## 2. Cellular uptake
### Cellular uptake of flavonoid glucuronides and their aglycones by OATP over-expressed cell lines

Three OATP over-expressed cell lines (HEK 293 OATP 1B1/1B3/2B1 over-expressed cell lines) were used in the uptake study because they are the most important hepatic uptake transporters for xenobiotics (Sun et al., 2013). Briefly, cells were seeded into 24-well plate 3 days before the experiment. Selected phenolic glucuronides were prepared in HBSS buffer (pH = 7.4) at required concentrations as working solution. Before incubation, cell culture medium was removed and cell was washed with 400 µl 37°C HBSS buffer (pH = 7.4) twice. Working solution was incubated with cells at 37°C for 20 min, when the intracellular concentrations reached steady state. After incubation, cells were washed with 400 µl ice-cold HBSS buffer (pH = 7.4) twice and cell pellet was collected in 200 µl of HBSS buffer. The cell pellet was further sonicated for 30 min. A total of 150 µl of suspensions was collected and 150 µl of acetonitrile (contain 0.2 µM rutin as internal standard) was added into the pellet suspension. The suspension was centrifuged at 15,000 rpm for 15 min and supernatant was collected for analysis to measure the intracellular concentration of glucuronides.

### Effect of specific inhibitors on the cell uptake

Fifty µM OATP inhibitors (rifampicin for OATP 1B1/1B3 and erlotinib or telmisartan for OATP 2B1) were prepared in working solutions. Working solutions without inhibitors were set as control group. Cells were incubated with their corresponding working solutions (containing different inhibitors) for 20 min.

### Effect of substrate concentration on the cell uptake

Selected phenolic glucuronides were prepared at 0.5, 1, 5, 10, 25, and 50 µM as working solutions. Cells were incubated with working solutions for 20 min. The $K_m$ and $V_{max}$ values were calculated by using the Michaelis-Menten equation.

### Cross inhibition on OATP1B1 cells using glucuronides and their corresponding aglycones

Several known substrates and non-substrates of OATP1B1 was applied as an inhibitor to inhibit the uptake of three selected substrates (Won-7-G, E2G and Lut-3'-G). Substrate concentration was 1 µM while inhibitor concentration was 25 µM. Cells incubated with substrate without inhibitor were set as control group (100% uptake). Relative uptake percentage was calculated by comparing intracellular concentrations of all experiment groups with the control group.

## 3. Portal vein infusion inhibition
### Determine the effect of OATP inhibition on the recycle of phenolic glucuronides

A cocktail consists of 1 mM rifampicin (OATP 1B1/1B3 inhibitor), 1 mM telmisartan (OATP 2B1 inhibitor), 1 mM E2G (OATP 1B1/1B3 substrate), and 1 mM E1S (OATP 2B1 substrate) were prepare in HBSS buffer (pH = 7.4) and used in the experiment. The experiment procedure was the same as we described earlier with minor modifications. To achieve a better inhibition effect, rats were infused with the inhibitor cocktail for 1 hr before the beginning of glucuronides infusion. After 1 hr of treatment, selected phenolic glucuronides were infused with inhibitor cocktail for 2.5 hr.

## 4. Small intestine perfusion
### Small intestine perfusion

To investigate the impact of intestinal metabolism on biliary excretion of phase II metabolites, small intestine perfusion study was performed as our previous publication with minor modifications (*Kemp et al., 2002*). Briefly, one segment of small intestine was perfused. The perfusion solution started from the beginning segment duodenum and went through 15 cm length of down-stream intestine after the inlet segment. Two µM of flavonoid aglycones (Won, Api, Bai) and phenolic drugs (Eze, Ral) were prepared together in HBSS buffer (pH = 7.4) as perfusion solution. The inlet cannulate was insulated and kept warm by a 37°C circulating water bath. The perfusion rate was set at 0.193 ml/min. The perfusion lasted for 2.5 hr. Bile samples and perfusate were collected. At the last time point, blood samples from tail vein and from hepatic portal vein were also collected and analyzed. The absorption (%) was calculated as below:

$$\text{Absorption}(\%) = \frac{\text{Perfused amount of aglycone} - \text{amount of aglycone in perfusate}}{\text{Perfused amount of aglycone}} \tag{A2-2}$$

## 5. Pharmacokinetic experiment
### Evaluation of LRE% on the pharmacokinetic profiles of phenolic compounds

To investigate whether the HER potentials have an impact on pharmacokinetic profiles of phenolic compounds, a pharmacokinetic experiment was conducted (n = 4). Briefly, Won and Bai were prepared together in oral suspension with ORA-Plus at the concentration of 50 mg/ml. Drugs were given to animals at the dose of 30 mg/kg. Eze/Ace were prepared and dosed the same way as described above for the other two flavonoids. Blood samples were taken at 0, 0.5, 1, 2, 4, 6, 8, and 24 hr after dosing. Samples were analyzed with LC/MS method after sample processing as detailed below.

**Appendix 2—table 1.** The hepatic expression level of OATP 1B1/1B3/2B1.
Expression level was presented as average ± SD. Data was obtained from Pubmed Gene database.

| Transporter | OATP 1B1[*] | OATP 1B1[†] | OATP 2B1[‡] |
| --- | --- | --- | --- |
| Expression (RPKM) | 119.3 ± 20.8 | 30.2 ± 3.9 | 44.0 ± 4.1 |
| Relative expression (%) | 61.7 | 15.6 | 22.7 |

*https://www.ncbi.nlm.nih.gov/gene/10599.

† https://www.ncbi.nlm.nih.gov/gene/28234.

‡ https://www.ncbi.nlm.nih.gov/gene/11309.

**Appendix 2—table 2.** The uptake of wogonin in four different cell lines.
The incubation concentration, time, condition and intracellular concentrations were summarized in the table. Inhibitor(s) were changed in different cell line. For OATP1B1 and OATP1B3 cell line, inhibitor was 50 µM rifampicin. For OATP 2B1 cell line, inhibitor was 50 µM telmisartan. For MDCK cell line, inhibitors were 50 µM rifampicin and 50 µM telmisartan. Student *t* test was applied to calculate the p values. No significant differences observed when wogonin was incubated with or without inhibitor.

| Cell type | Incubation concentration(µM) | Incubation time (min) | Inhibitor | Intracellular concentration(nM) |
| --- | --- | --- | --- | --- |
| MDCK MRP3 over-expressed cell line | 5 | 120 | Yes | 150.33 ± 37.87 |
| | 5 | 120 | No | 163.67 ± 21.57 |
| HEK-293 OATP1B1 over-expressed cell line | 10 | 20 | Yes | 442.00 ± 309.56 |
| | 10 | 20 | No | 365.67 ± 168.67 |
| HEK-293 OATP1B3 over-expressed cell line | 10 | 60 | Yes | 333.33 ± 33.08 |
| | 10 | 60 | No | 381.33 ± 128.82 |

*Continued on next page*

*Appendix 2—table 2 continued*

| Cell type | Incubation concentration(μM) | Incubation time (min) | Inhibitor | Intracellular concentration(nM) |
|---|---|---|---|---|
| HEK-293 OATP2B1 over-expressed cell line | 10 | 20 | Yes | 782.33 ± 115.42 |
| | 10 | 20 | No | 897.00 ± 409.17 |

**Appendix 2—table 3.** The uptake of wogonin and wogonin-7-G (wogonoside) in three different cell lines.

The incubation concentration, time, condition and intracellular concentrations were summarized iin the table. Inhibitor(s) were changed in different cell line. For OATP1B1 and OATP1B3 cell line, inhibitor was 50 μM rifampicin. For OATP 2B1 cell line, inhibitor was 50 μM telmisartan. Student *t* test was applied to calculate the p values. Significant differences were observed when wogonoside incubated with or without inhibitors in OATP 1B1 and 1B3 cell lines. No significant differences observed when wogonin was incubated with or without inhibitor.

| Cell type | Incubation concentration (μM) | Incubation time (min) | Inhibitor | Wogonoside intracellular concentration (nM) | Wogonin intracellular concentration (nM) |
|---|---|---|---|---|---|
| HEK-293 OATP1B1 over-expressed cell line | 10 | 20 | Yes | 0.01 ± 0.01 | 442.00 ± 309.56 |
| | 10 | 20 | No | 401.00 ± 29.60†† | 365.67 ± 168.67 |
| HEK-293 OATP1B3 over-expressed cell line | 10 | 60 | Yes | 48.33 ± 9.12 | 333.33 ± 33.08 |
| | 10 | 60 | No | 58.03 ± 16.60 | 381.33 ± 128.82 |
| HEK-293 OATP2B1 over-expressed cell line | 10 | 20 | Yes | 34.57 ± 21.26 | 782.33 ± 115.42 |
| | 10 | 20 | No | 82.87 ± 2.45* | 897.00 ± 409.17 |

*p< 0.05.
†p<0.01.

**Appendix 2—table 4.** The liver concentration of wogonin/wogonin-7-G (n = 3).

| Compound | Wogonin | Wogonin-7-G (Wogonoside) |
|---|---|---|
| Liver concentration (nmol/g) | 0.51 ± 0.28 | <0.1* |
| Ratio (Wogonin/Wogonoside) | 5 > | \ |

* The concentration was below quantification limit (4 nM).

**Appendix 2—table 5.** Biliary glucuronide secretion rate and liver recycling efficiency of 16 different glucuronides.

| Infused compound (10 μM) | Secretion rate (nmol/hr)* | LRE%† |
|---|---|---|
| Ezetimibe-4'-G | 19.31 ± 1.85 | 96.5 ± 9.2 |
| Wogonin-7-G(wogonoside) | 16.3 ± 4.28 | 81.5 ± 21.4 |
| Raloxifene-4'-G | 11.48 ± 2.32 | 57.4 ± 11.6 |
| Apigenin-7-G | 10.64 ± 4.49 | 53.2 ± 22.4 |
| Raloxifene-6-G | 10.53 ± 1.51 | 52.6 ± 7.5 |
| Chrysin-7-G | 9.91 ± 1.92 ‡ | 49.6 ± 9.6 ‡ |
| Genestein-7-G | 9.45 ± 3.21 ‡ | 47.3 ± 16.0 ‡ |
| Icaritin-7-G | 7.46 ± 0.72 | 37.3 ± 3.6 |
| Icaritin-3-G | 5.35 ± 1.12 | 26.8 ± 5.6 |

*Continued on next page*

*Appendix 2—table 5 continued*

| Infused compound (10 µM) | Secretion rate (nmol/hr)* | LRE%† |
|---|---|---|
| Scuttelarin | 4.34 ± 1.18 | 21.7 ± 5.9 |
| Biochanin A-G | 3.48 ± 0.75 ‡ | 17.5 ± 3.7 ‡ |
| Luteolin-3'-G | 1.33 ± 0.34 | 6.7 ± 1.7 |
| Baicalin | 0.69 ± 0.30 | 3.4 ± 1.5 |
| Luteolin-7-glycoside | 0.67 ± 0.56 | 3.4 ± 2.8 |
| Quercetin-3-G | 0.43 ± 0.57 | 2.2 ± 2.8 |
| Ace-G | 0.10 ± 0.06 | 0.5 ± 0.3 |

*The glucuronide secretion rate at the steady state was calculated by the linear regression of accumulated amount secreted in bile vs. time.

† LRE (liver recycling efficiency) % = Excretion rate at steady state/infusion rate.

‡ Data from previous published study (*Zeng et al., 2016*).

**Appendix 2—table 6.** Gender differences in bile secretion rates and liver recycling efficiency (LRE %).

| Infusion compounds (10 µM) | Secretion rate (nmol/hr) | | LRE | (%) |
|---|---|---|---|---|
| | Male | Female | Male | Female |
| Won-7-G | 16.30 ± 4.28 | 15.06 ± 0.70 | 81.5 ± 21.4 | 75.3 ± 3.5 |
| Bai-7-G | 0.69 ± 0.29 | 1.78 ± 0.42 | 3.4 ± 1.5 | 8.9 ± 2.1 |
| Lut-3'-G | 1.33 ± 0.34 | 1.67 ± 0.32 | 6.7 ± 1.7 | 8.3 ± 1.6 |

**Appendix 2—table 7.** Kinetic parameters of uptake (Km and Vmax) of selected glucuronides wogonoside, luteolin-3'-glucuronide (Lut-3'-glu) and apigenin-7-glucuronide (Api-7-glu) in OATP1B1/1B3/2B1 over-expressed cells.

| | 1B1 | | 1B3 | | 2B1 | |
|---|---|---|---|---|---|---|
| Compound | Km(µM) | Vmax (pmol/min) | Km(µM) | Vmax (pmol/min) | Km(µM) | Vmax (pmol/min) |
| Won-7-G | 27.68 | 53.37 | >50 | 11.72 | >50 | 55.26 |
| Lut-3'-G | 4.46 | 22.79 | >50 | 9.62 | 0.89 | 4.07 |
| Api-7-G | >50 | >200 | >50 | >200 | 9.72 | 5.09 |
| Eze-4'-G | >50 | 11.45 | 17.64 | 1.67 | 8.55 | 1.61 |

## Appendix 3

### Bioanalysis methods

Methods applied in the analysis of biological samples were summarized in this section. The sample process methods and parameters of LC/MS methods were also presented below.

### 1.Sample process

#### Blood

A volume of 10 µl blood was mixed with 10 µl of water and 200 µl of acetonitrile (ACN) (with 0.2 µM rutin as internal standard) was added into the mixture. After vortexing for 1 min, the mixture was centrifuged at 15,000 rpm for 15 min. The supernatant was dried under air at room temperature and reconstitute with 100 µl of 20% ACN for analysis. Standard samples of known concentrations were prepared in the same way.

#### Bile

A volume of 5 µl bile was mixed with 5 µl of water and further dilute into 1 ml of mixture (containing 0.1 µM rutin as internal standard). The mixture was further loaded on an OasisR HLB SPE cartridge and eluted by 1 ml of water, 3 ml of 40% methanol. One milliliter of MeOH was used as final elution fluid and sample was dried under air blowing at room temperature after elution. The residue was reconstituted with 200 µl 20% ACN for analysis.

#### Cells

Cell samples were processed by adding 150 µl ACN (containing 0.2 µM rutin as internal standard) into 150 µl of cell pellet. After centrifuge at 15,000 rpm for 15 min, the supernatant was collected for analysis.

#### Perfusate

A volume of 50 µl perfusate sample was mixed with 50 µl of water and 100 µl of ACN was (with 0.2 µM rutin as internal standard) added into the mixture. After vortex for 1 min, the mixture was centrifuged at 15,000 rpm for 15 min. 100 µl of supernatant was taken for analysis.

### 2. LC/MS methods

#### UPLC conditions

UPLC condition: system, Waters Acquity with diode array; column, Restek Ultra BPh (5 µm, 100 mm × 2.1 mm); mobile phase, A 0.1% formic acid in water, mobile phase B acetonitrile; gradient, 10% B (0–0.5 min), 10% B-34% B (0.5–1.0 min), 34% B-60% B (1.0–2.5 min), 60% B-95% B (2.5–6.0 min), 95% B-10% B (6.0–7.0 min); flow rate,0.45 ml/min; column temperature,45 ˚C; injection volume, 10 µL.

#### Mass spectrometry conditions

The MS analysis was performed on a Sciex 5500 triple quadrupole mass spectrometer (AB Sciex LLC, Framingham, MA) equipped with an ESI source. The detection was conducted using MRM scan type in positive ion mode. The instrument dependent parameters were: ion-spray voltage, 5.5 kV; ion source temperature, 400˚C; nebulizer gas (gas 1), nitrogen, 20 psi; turbo gas (gas 2), nitrogen 20 psi; curtain gas, nitrogen 20 psi. Unit mass resolution was set in both mass-resolving quadruples Q1 and Q3. The compound dependent parameters were summarized in *Appendix 3—table 1*.

**Appendix 3—table 1.** Compound-dependent parameter of the analytes and I.S.

| Compound | Q1/Q3 | DP | CE | EP | CXP |
|---|---|---|---|---|---|

*Continued on next page*

*Appendix 3—table 1 continued*

| Compound | Q1/Q3 | DP | CE | EP | CXP |
|---|---|---|---|---|---|
| Api-7-G | 445.0/269.0 | −90 | −34 | −10 | −23 |
| Ace-G | 326.0/150.0 | −100 | −34 | −10 | −15 |
| Bai-7-G | 445.0/269.0 | −120 | −30 | −10 | −19 |
| E1S | 349.0/269.0 | −45 | −52 | −10 | −19 |
| E2G | 447.0/271.0 | −170 | −40 | −10 | −19 |
| Eze-4'-G | 584.0/271.0 | −154 | −42 | −10 | −15 |
| Ica-3-G | 543.1/367.1 | −90 | −33 | −10 | −10 |
| Ica-7-G | 543.1/352.0 | −90 | −53 | −10 | −10 |
| Lut-3'-G | 461.2/285.2 | −87 | −29 | −10 | −15 |
| Que-3-G | 477.0/301.0 | −110 | −32 | −10 | −11 |
| Ral-4'-G | 649.3/473.3 | −80 | −44 | −10 | −13 |
| Ral-6-G | 649.3/473.3 | −80 | −44 | −10 | −13 |
| Scu-7-G | 461.0/285.0 | −70 | −34 | −10 | −13 |
| Won-7-G | 459.0/283.0 | −87 | −21 | −10 | −15 |
| Rutin (I.S.) | 609.0/300.0 | −87 | −54 | −10 | −21 |

## Blood stability test

### Blood stability test of Won-7-G, Api-7-G, and Bai-7-G

The stability of Won-7-G, Api-7-G, and Bai-7-G in blood matrix were evaluated by adding standard compounds into rat blood. Stock solutions of glucuronides were added into 1 ml of rat blood to reach the concentrations at 2, 10, and 25 µM (n = 3). Samples were further incubated in water bath at 37°C. Fifty µl of samples were collected every 0.5 hr. The incubation lasted for 2.5 hr. Samples collected at 0 hr point were set as control (100%, no degradation). Samples were further analyzed after processing as described above. Relative stability was calculated by comparing blood concentrations at each sampling time point to blood concentration at the '0 hr' time point. The results of blood stability test could be found in *Figure 2—figure supplement 3*.

## Appendix 4

## Fisher exact test of correlation analysis

### 1.Intracellular concentration

The intracellular concentrations were calculated by adding the uptake results in 3 OATP isoforms (OATP1B1, 1B3 and 2B1) together with weightings. The intracellular concentrations were calculated as the equation presented below:

$$Total intracellular concentration =$$

$$\frac{X}{10}*OATP1B1 concentration + \frac{Y}{10}*OATP1B3 concentration + \frac{Z}{10}*OATP2B1 concentration$$

Concentrations are intracellular concentration of phenolic glucuronide in 3 cell lines generated from cell uptake experiment. X, Y, Z are integers and are weightings of individual isoform, which follows the description below:

X + Y + Z = 10, $0 \leq X \leq 10$, $0 \leq Y \leq 10$, $0 \leq Z \leq 10$.

Different X, Y, and Z numbers indicated different relative expression levels of individual OATP isoform. It also describes the contribution of specific isoform in the hepatic uptake of the substrates.

### 2. Fisher exact test

The liver recycling efficiency (LRE%) obtained from rat infusion experiment were correlated with the intracellular concentration obtained from cell uptake results. Since cell studies were conducted using human OATPs over-expressed cell lines, it was important to determine if there is a statistical correlation between these two parameters by using Fisher exact test (*Kosoglou et al., 2005*).

To conduct Fisher exact test, all compounds were classified into four groups by their LRE% (high and low) and intracellular concentrations (high and low). LRE% higher than 50% are considered 'high recycling' and concentrations higher than the average values are considered 'high uptake'. The same definition was applied in 'low recycling' and 'low uptake'.

The number of substrates that belongs to the classified groups (e.g. high recycle and high uptake, high recycle and low uptake) was counted. A 2*two table was generated from the classification. An example of the classification was presented below:

Then Fisher exact test was applied to test the statistically significant correlation of two parameters (liver recycling efficiency and intracellular concentration) influencing the distribution of the classified numbers. $p<0.05$ was considered of significant correlation. Different X, Y, Z weighting combinations were assigned and individual p values were calculated (*Appendix 4—table 2*).

OATP 1B1 is the most abundant uptake transporter expressed in liver (Appendix 2 table S4). It is also the most important uptake transporter due to its substrate spectrum. OATP 1B3 is grouped with OATP1B1 in both localization and substrates, but the contribution of OATP 1B31 is smaller than OATP 1B1. Therefore, combinations which assigned the highest weightings to OATP1B3 were considered not physiologically relevant and eliminated from the results.

**Appendix 4—table 1.** Example of fisher exact test generated by the methods described above.

|  |  | Liver recycling efficiency (%) | |
|---|---|---|---|
|  |  | **High** | **Low** |
| Intracellular concentrations (nM) | High | 4 | 1 |
|  | Low | 0 | 4 |

**Appendix 4—table 2.** The results of Fisher exact test in different weightings were summarized.
p Values of different OATP weightings were calculated and summarized in this table. p<0.05 indicated significant correlation found between recycle ratios and cell uptake. Results marked in red were eliminated.

| Weighting | | | p | Weighting | | | p |
|---|---|---|---|---|---|---|---|
| OATP1B1 | OATP1B3 | OATP2B1 | value | OATP1B1 | OATP1B3 | OATP2B1 | value |
| 2 | 3 | 5 | 0.0079 | 7 | 1 | 2 | 0.0476 |
| 2 | 2 | 6 | 0.0079 | 7 | 0 | 3 | 0.0476 |
| 2 | 1 | 7 | 0.0079 | 8 | 2 | 0 | 0.0476 |
| 2 | 0 | 8 | 0.0079 | 8 | 0 | 2 | 0.0476 |
| 3 | 2 | 5 | 0.0079 | 8 | 1 | 1 | 0.0476 |
| 3 | 1 | 6 | 0.0079 | 9 | 1 | 0 | 0.0476 |
| 3 | 0 | 7 | 0.0079 | 9 | 0 | 1 | 0.0476 |
| 4 | 1 | 5 | 0.0079 | 10 | 0 | 0 | 0.0476 |
| 4 | 0 | 6 | 0.0079 | 0 | 10 | 0 | 0.1667 |
| 5 | 1 | 4 | 0.0079 | 0 | 9 | 1 | 0.1667 |
| 5 | 0 | 5 | 0.0079 | 0 | 8 | 2 | 0.1667 |
| 6 | 0 | 4 | 0.0079 | 0 | 7 | 3 | 0.1667 |
| 0 | 3 | 7 | 0.0476 | 0 | 6 | 4 | 0.1667 |
| 0 | 2 | 8 | 0.0476 | 0 | 5 | 5 | 0.1667 |
| 0 | 1 | 9 | 0.0476 | 0 | 4 | 6 | 0.1667 |
| 0 | 0 | 10 | 0.0476 | 1 | 9 | 0 | 0.1667 |
| 1 | 3 | 6 | 0.0476 | 1 | 8 | 1 | 0.1667 |
| 1 | 2 | 7 | 0.0476 | 1 | 7 | 2 | 0.1667 |
| 1 | 1 | 8 | 0.0476 | 1 | 6 | 3 | 0.1667 |
| 1 | 0 | 9 | 0.0476 | 1 | 5 | 4 | 0.1667 |
| 3 | 3 | 4 | 0.0476 | 1 | 4 | 5 | 0.1667 |
| 4 | 4 | 2 | 0.0476 | 2 | 8 | 0 | 0.1667 |
| 4 | 3 | 3 | 0.0476 | 2 | 7 | 1 | 0.1667 |
| 4 | 2 | 4 | 0.0476 | 2 | 6 | 2 | 0.1667 |
| 5 | 4 | 1 | 0.0476 | 2 | 5 | 3 | 0.1667 |
| 5 | 3 | 2 | 0.0476 | 2 | 4 | 4 | 0.1667 |
| 5 | 2 | 3 | 0.0476 | 3 | 7 | 0 | 0.1667 |
| 6 | 4 | 0 | 0.0476 | 3 | 6 | 1 | 0.1667 |
| 6 | 3 | 1 | 0.0476 | 3 | 5 | 2 | 0.1667 |
| 6 | 2 | 2 | 0.0476 | 3 | 4 | 3 | 0.1667 |
| 6 | 1 | 3 | 0.0476 | 4 | 6 | 0 | 0.1667 |
| 7 | 3 | 0 | 0.0476 | 4 | 5 | 1 | 0.1667 |
| 7 | 2 | 1 | 0.0476 | 5 | 5 | 0 | 0.1667 |

## Appendix 5

### Microsome incubation

1.Microsome incubation

40 µM L3'G is incubated with male F344 rat liver microsome (final concentration of 0.053 mg of protein per mL mixture) for 2 hr. The reaction is stopped by adding acetonitrile of same volume of the mixture. By analyzing the microsome samples with UPLC and a UV detector, a peak is observed at 1.79 min with similar bands in UV spectra (*Figure 1*.B1 and 1.B2). We presume the peak consists of one or more di-glucuronide(s) of luteolin. To confirm the peak is glucuronide metabolite(s) of L3'G, we performed hydrolysis study with beta-glucuronidase, which converts glucuronides into aglycones.

2.Hydrolysis study of microsome samples

Samples from microsome study are incubated with beta-glucuronidase (at a final concentration of 900 unit/mL) for 2 hr. Both the postulated di-glucuronide peak and L3'G peak disappeared while a luteolin peak appear at 3.02 min (*Figure 2*.C1 and 2 .C2).

To further investigate the hydrolysis process, a separate time-dependent hydrolysis study is performed. A final concentration of 180 unit/mL beta-glucuronidase is added in 100 µL microsome samples. The reaction is stopped at 5 min and 30 min. From the UPLC-UV results, the postulated di-glucuronide peak and L3'G peak both decrease with time, and the luteolin peak increases with time (*Figure 2*.A1 and 2.B1).

Based on the microsome incubation results, we confirm that luteolin 3'-glucuronide can be further metabolized in rat liver microsome.

**Appendix 5—table 1.** Peak area of postulated di-glucuronide of luteolin, L3'G, and luteolin from UPLC chromatograph (n = 2).

|  | Retention time/min | Before hydrolysis | 5 min hydrolysis | 30 min hydrolysis |
|---|---|---|---|---|
| Postulated di-glucuronide | 1.79 | 28543 | 26070.5 | 18922 |
| L3'G | 2.67 | 78905.5 | 69006 | 42460.5 |
| Luteolin | 3.02 | - | 17289 | 66341 |
| sum |  | 107448.5 | 112365.5 | 127723.5 |

