## [Decision Letter]

**Acceptance summary:**

Tu and colleagues present an interesting body of work describing what they feel is a novel method of recycling of glucuronidated metabolites, specifically those of phenolic compounds. Rather than relying on the traditionally held pathway wherein a compound is metabolized in the liver and then recycled in the intestine after enzymes of the gut microbiota remove the glucuronide (enterohepatic recirculation/recycling – EHR), this mechanism involves glucuronidation of the compound in the small intestine, uptake of the glucuronidated metabolites into the liver by OATP1 transporters and then transport back out of the liver into the bile and subsequently into the small intestine where the glucuronide is recycled into its aglycone form by gut microbiota enzymes (hepatoenteric recycling -HER).

**Decision letter after peer review:**

Thank you for submitting your article "Hepatoenteric Recycling Is A New Disposition Mechanism for Orally Administered Phenolic Drugs and Phytochemicals" for consideration by *eLife*. Your article has been reviewed by 2 peer reviewers, and the evaluation has been overseen by a Reviewing Editor and Olga Boudker as the Senior Editor. The following individual involved in review of your submission has agreed to reveal their identity: Noelle S Williams (Reviewer #2).

The reviewers have discussed the reviews with one another and the Reviewing Editor has drafted this decision to help you prepare a revised submission.

Summary:

Tu and colleagues present a body of work describing a novel method of recycling of glucuronidated metabolites, specifically those of phenolic compounds. Rather than relying on the traditionally held pathway wherein a compound is metabolized in the liver and then recycled in the intestine after enzymes of the gut microbiota remove the glucuronide (enterohepatic recirculation/recycling – EHR), this mechanism involves glucuronidation of the compound in the small intestine, uptake of the glucuronidated metabolites into the liver by OATP1 transporters and then transport back out of the liver into the bile and subsequently into the small intestine where the glucuronide is recycled into its aglycone form by gut microbiota enzymes (hepatoenteric recycling -HER). The authors argue that this has relevance for drug-drug interactions involving the OATP transporters and as a means to shuttle glucuronidated metabolites to the colon since the metabolism of phenolics by intestinal cells to glucuronidated forms as well as subsequent recycling means there should be large amounts of these metabolites in the colon.

Essential revisions:

– The introduction could be more directed to the general audience.

– In describing Figure 2A4, the authors claim that RR% is plotted against steady state blood concentration which "…displayed a saturating trend, indicative of transporter-mediated excretion." First, excretion rate not RR% is plotted and although these values tend to trend together the authors should be more precise in their description and better explain their rationale for making the claim that this indicates transport. It would also be helpful if there were more data in between 100 and 1000 uM.

– In Figure 2B1-7, the authors show decreased recycling of the glucuronide relative to the aglycone form by monitoring appearance of the glucuronide in the bile after hepatic infusion. While the difference is dramatic, it is hard to know the relative roles of metabolism of the aglycone to the glucuronide versus uptake and transport of the aglycone form. It might be helpful to evaluate levels of the aglycone and glucuronide inside the liver after this experiment.

– The authors say they are examining the role of protein binding in section 3.1.3 by evaluating a higher concentration of Won aglycone, making the assumption that this would saturate protein binding. This seems somewhat arbitrary and if they truly wanted to assess the role of saturable protein binding, should do a protein binding experiment. It seems it would have been better to directly evaluate uptake of the aglycone form as previously suggested and just eliminate this evaluation.

– In section 3.1.4, the authors evaluate different flavonoids to make a comment on the specific role of glucuronidation at the 7-position in RR%. This doesn't make sense. One should evaluate the same compound that is glucuronidated at multiple sites to evaluate the role of glucuronidation location – and correlate this with cell based studies with the transporters. It was also unclear why the 7 location was so relevant. The authors should elaborate.

– The experiments describing sex differences and cassette dosing on glucuronide RR% seems insufficient and unnecessary for this manuscript.

– In section 3.2.1 when discussing the outcome for Lut-3'-G it would be helpful if the authors could refer to the data for this point in figure 2B4 and to show the evidence for additional metabolism in supplementary data.

– In Section 3.2.4 where the authors make the claim that substrates of the transporters make better inhibitors, it would again be helpful to evaluate uptake of the aglycone forms rather than just looking at accumulation of glucuronides.

– The authors need to elaborate further in section 3.4 on the lack of physiological relevance for the 18/25 examples where there was no correlation between cellular uptake and RR% and the 7 which were not relevant with respect to relative expression of OATPs in human liver. Does the latter have something to do with Table S4? If so, that table should be referenced here.

– Please describe the relevance of the blood stability test. This seems something included to satisfy a previous reviewer without any reference to the relevance of the finding to the authors' overall conclusion.

[Editors' note: further revisions were suggested prior to acceptance, as described below.]

Thank you for resubmitting your work entitled "Hepatoenteric Recycling Is A New Disposition Mechanism for Orally Administered Phenolic Drugs and Phytochemicals in Rat" for further consideration by *eLife*. Your revised article has been evaluated by Olga Boudker (Senior Editor) and a Reviewing Editor.

Tu and colleagues present an interesting body of work describing what they feel is a novel method of recycling of glucuronidated metabolites, specifically those of phenolic compounds. Rather than relying on the traditionally held pathway wherein a compound is metabolized in the liver and then recycled in the intestine after enzymes of the gut microbiota remove the glucuronide (enterohepatic recirculation/recycling – EHR), this mechanism involves glucuronidation of the compound in the small intestine, uptake of the glucuronidated metabolites into the liver by OATP1 transporters and then transport back out of the liver into the bile and subsequently into the small intestine where the glucuronide is recycled into its aglycone form by gut microbiota enzymes (hepatoenteric recycling -HER).

The manuscript has been improved but there are some remaining issues that need to be addressed, as outlined below:

Essential revisions:

1) Please provide a reference for the statement in the last sentence of the introduction on line 135.

2) Please define meaning of "LRE%" abbreviation upon its first use on line 140.

3) Please change recycle rate to LRE% on line 554.

---

## [Author Response]

Summary:Tu and colleagues present a body of work describing a novel method of recycling of glucuronidated metabolites, specifically those of phenolic compounds. Rather than relying on the traditionally held pathway wherein a compound is metabolized in the liver and then recycled in the intestine after enzymes of the gut microbiota remove the glucuronide (enterohepatic recirculation/recycling – EHR), this mechanism involves glucuronidation of the compound in the small intestine, uptake of the glucuronidated metabolites into the liver by OATP1 transporters and then transport back out of the liver into the bile and subsequently into the small intestine where the glucuronide is recycled into its aglycone form by gut microbiota enzymes (hepatoenteric recycling -HER). The authors argue that this has relevance for drug-drug interactions involving the OATP transporters and as a means to shuttle glucuronidated metabolites to the colon since the metabolism of phenolics by intestinal cells to glucuronidated forms as well as subsequent recycling means there should be large amounts of these metabolites in the colon.

In addition to the responses to the reviewers’ comments, we also made several changes in the terminology to better clarify the concept of liver recycling efficiency, recirculation and recycling. For example, we change the recirculation ratio (RR%) to liver recycling efficiency (LRE%), since the results we calculated did not actually reflect the whole process of HER. Only the recycling ratio of liver was determined. In addition, “recirculation” has been replaced by “recycling” at several places of the manuscript.

Lastly, because editors and reviewers were concerned about making our topic more general and as such we inserted the following paragraph into the “Introduction” to better address this concern.

“EHR differs from HER in the organs for formation and disposition of phase II metabolites. We believe It is important to delineate HER from EHR since the formation and disposition of phase II metabolites could be greatly different in disease condition related to different organs that could lead to differences in drug efficacy and toxicity both in systemic circulation and intestinal lumen. We believe that the differences in phase II enzymes, efflux and uptake transporters in different metabolic organs (such as liver and intestine) based on gender and disease conditions plays a significant role in determining the impact of EHR vs HER on the drug disposition.”

Essential revisions:– The introduction could be more directed to the general audience.

We have modified the introduction as suggested by the reviewers. The contents are now made more directed to general introduction. (see above)

We have added the clinical relevance for understanding the difference between EHR and HER recycling processes with references. (see above)

We also got the feedback from colleagues not in the same area of research on the introduction and incorporate their comments in the introduction.

– In describing Figure 2A4, the authors claim that RR% is plotted against steady state blood concentration which "…displayed a saturating trend, indicative of transporter-mediated excretion." First, excretion rate not RR% is plotted and although these values tend to trend together the authors should be more precise in their description and better explain their rationale for making the claim that this indicates transport. It would also be helpful if there were more data in between 100 and 1000 uM.

The reviewer is correct that we did not plot LRE% against the steady-state blood concentration. Instead we plotted the excretion rates against steady-state blood concentration. The saturation in the excretion rate as a function of blood concentration is expected if the blood concentration is correlated with the biliary excretion of wogonoside. This means that a higher blood concentration will result in a higher rate of excretion until the transporter is saturated. We also agreed that having more concentrations between 100 and 1000 μM would be valuable, but that was not done. However, the current data are enough to show that the transporter is involved because process is saturable.

Another reason we believe that the uptake/efflux transporters are involved in the recycling process is because of the hydrophilicity of glucuronides (precluding significant passive diffusion). The following is the revised changes along with page and paragraph information.

Page 17, paragraph 2:

“It showed that at lower blood concentrations (<100μM) of wogonoside, its biliary excretion rates increased linearly. At higher blood concentrations (>=100μM), biliary excretion rates showed a saturation trend, indicative of transporter-mediated excretion (Figure 2A4).”

– In Figure 2B1-7, the authors show decreased recycling of the glucuronide relative to the aglycone form by monitoring appearance of the glucuronide in the bile after hepatic infusion. While the difference is dramatic, it is hard to know the relative roles of metabolism of the aglycone to the glucuronide versus uptake and transport of the aglycone form. It might be helpful to evaluate levels of the aglycone and glucuronide inside the liver after this experiment.

We do not disagree with the suggestion that “It might be helpful to evaluate levels of the aglycone and glucuronide inside the liver after this experiment”. Our recent measurement of liver concentration of Wogonin and Wogonoside showed a high ratio (> 5) of aglycone/glucuronide (Appendix 2 Table S4), providing evidence that metabolism of aglycone is slow. Furthermore, our uptake studies in the several OATP-overexpressed cell lines (Appendix 2 Table S2 and table S3) indicated that uptake/transport of aglycone was always equal to or faster than corresponding glucuronides. Also, there was no significant reduction in the uptake of Wogonin in the presence of OATP inhibitors, suggesting that the hepatic uptake of aglycone is primarily by passive diffusion and not active uptake. Lastly, in Caco-2 cells studies, flavonoid aglycones always displayed high permeability [1-3]. Therefore, there is no evidence that the hepatic uptake of phenolic aglycones would be slow and become a rate-limiting step in the biliary excretion of glucuronides, suggesting that a low LRE% for the aglycones meant slow metabolism in the liver rather than slow uptake of aglycones.

The following is the revised changes along with page and paragraph information.

Page 18, paragraph 2:

“Liver tissue concentrations of wogonin was at least 5 times higher than won-7-G concentrations (Appendix 2 supplement Table S4), which indicated that slow metabolism in liver was the likely reason for the lower LRE%.”

Page 18, paragraph 1:

“To rule out the possibility that slow aglycone uptake was the reason for lower LRE%, an uptake comparison between Won and Won-7-G in 3 cell lines were also conducted (Appendix 2 supplement Table S2 and Table S3). The results indicated that the uptake of Won was not influenced in the presence of specific transporter inhibitor, while Won-7-G was greatly inhibited, suggesting that aglycone uptake was mainly by passive diffusion, and often faster than their corresponding glucuronides.”

– The authors say they are examining the role of protein binding in section 3.1.3 by evaluating a higher concentration of Won aglycone, making the assumption that this would saturate protein binding. This seems somewhat arbitrary and if they truly wanted to assess the role of saturable protein binding, should do a protein binding experiment. It seems it would have been better to directly evaluate uptake of the aglycone form as previously suggested and just eliminate this evaluation.

As stated in the answer to the question #3, there is no evidence to suggest that uptake of flavonoid aglycones is slow, and therefore become the rate-limiting step in hepatic glucuronide formation. Based on the uptake experiments performed in our lab, we believe that uptake of aglycones into the hepatocytes is equal to or faster than the corresponding glucuronides. Therefore, apart from the slow metabolism, another possible explanation for low LRE% could be the plasma protein binding of phenolic aglycone, which impedes its hepatic transport by decreasing the effective concentration available to the liver.

We hypothesize that if protein binding plays a significant role, LRE% would be higher at a higher aglycone concentration once the protein binding becomes saturated. But our results showed the opposite, indicating that protein binding was not the major reason for low LRE%. Therefore, we are quite certain that decrease in LRE% is not the result of protein binding but possibly saturation of metabolic enzymes. This is supported by the ratio of liver concentration of wogonin and wogonoside (Table S1) when aglycone is infused in portal vein. It indicated that the decrease of LRE% was due to the slow metabolism.

The following is the revised changes along with page and paragraph information.

Page 18, paragraph 2:

“However, the results indicated that LRE% at a higher concentration (100μM) was significantly lower (*P<.001*) than at a lower concentration (2μM) (Figure 2C3), ruling out the impact of extensive protein binding on the lower LRE% of aglycones. Liver tissue concentrations of wogonin was at least 5 times higher than won-7-G concentrations (Appendix 2 supplement Table S4), which indicated that the slow metabolism in liver was the likely reason for the lower LRE%.”

– In section 3.1.4, the authors evaluate different flavonoids to make a comment on the specific role of glucuronidation at the 7-position in RR%. This doesn't make sense. One should evaluate the same compound that is glucuronidated at multiple sites to evaluate the role of glucuronidation location – and correlate this with cell based studies with the transporters. It was also unclear why the 7 location was so relevant. The authors should elaborate.

The 7-position is highly relevant because major dietary important flavonoids have a hydroxyl group at 7 position, and several UGTs are shown to have region-selective glucuronidation at this position [4]. For flavonoid compounds such as genistein [5], chrysin [6], apigenin [7], wogonin [8], and baicalein [9], at least one of their main glucuronides are conjugated at the 7position. We agreed with the reviewer that it would be better to study this problem with a flavonoids that have 3-5 OH groups and then make synthetic glucuronide for each of these OH positions, but this was not scope of the current study as it requires huge amount of efforts to purify mono-glucuronides from a mixture of other mono-glucuronides. Multiple monoglucuronides are expected if we use biosynthesis method, and chemical synthesis of these mono-glucuronides is beyond the scope of this work. However, we have turned down the importance of this 7-position in our discussion.

– The experiments describing sex differences and cassette dosing on glucuronide RR% seems insufficient and unnecessary for this manuscript.

For sex differences:

OATPs were identified as the major transporters determining LRE%. Since Oatps were reported to be male-predominant in rat liver [10], we hypothesized that sex differences could cause the changes of OATP expression levels and required investigating. In addition, E2G, a precursor for female sex hormone and a classical OATP substrate, has a drastically different blood concentrations in male and female. Therefore, we tested some of our model compounds to verify whether the sex difference matters.

For cassette dosing:

It’s a labor intensive and animal intensive experiments to determine LRE% of glucuronides. In order to improve the throughput of the experimental determination of LRE% and save the number of animal usage (IACUC 3R’s principle), we infused multiple compounds simultaneously. We chose to use a substrate concentration of 10μM because at concentrations lower than 10μM, many glucuronides cannot be detected in bile samples. Whereas experiment at higher concentration requires a much large quantities of glucuronides, which are generally not commercially available and hard to obtain via biosynthesis.

Another reason we did not use higher concentration because it is possible that these glucuronides would have competitive inhibition when infused together. To rule out the possibility that they may still compete at 10μM, we compared the single and cassette dosing of two model compounds (high recycle and low recycle compounds) and found no significant differences between single compound infusion and cassette infusion.

The following is the revised changes along with page and paragraph information.

Page 19, paragraph 2:

“We are interested in determining the effects of sex difference on the OATP liver expression levels, because the expression level of Oatp in rat liver was reported to be significantly higher in males than in females [20]. Also, E2G is a classical OATP substrate that has a drastically higher blood concentrations in females than males. The results indicated that there was not sex related differences in the LRE% of Won-7-G and Lut-3’-G. (Appendix 2 supplement Table S6).

Interestingly, the LRE% of Bai-7-G increased from ~4% (male rats) to ~10% (female rats) *(P<.05)*.”

– In section 3.2.1 when discussing the outcome for Lut-3'-G it would be helpful if the authors could refer to the data for this point in figure 2B4 and to show the evidence for additional metabolism in supplementary data.

As per reviewer’s suggestion, we have added additional reference in section 3.2.1. We have also added additional evidence of metabolite formation using in vitro glucuronidation of Lut-3'-G in rat liver microsomes (Figure 2 supplement Figure S1A-S1C) that may explain the low LRE% of Lut-3’-G. Our results showed that Lut-3’-G could be further metabolized to Lut-di-G (di-glucuronides), which may provide an explanation why Lut-3’-G was present in bile at much lower than expected concentration. Luteolin was previously reported to be metabolized into Lut-di-G, which is consistent with our observation [11].

The following is the revised changes along with page and paragraph information.

Page 20, paragraph 3:

“The discrepancy can be explained by the fact that Lut-3’-G could be further metabolized into diglucuronides of Lut in vivo [23]. A liver microsome-mediated metabolism of Lut-3’-G confirmed the formation of Lut di-glucuronides by LC-MS (Figure 2 supplement Figure S2A and Figure S2B). Moreover, the peaks of Lut-3’-G and Lut-di-G both decreased upon incubation with glucuronidases (Figure 2 supplement Figure S2C).”

– In Section 3.2.4 where the authors make the claim that substrates of the transporters make better inhibitors, it would again be helpful to evaluate uptake of the aglycone forms rather than just looking at accumulation of glucuronides.

We agreed that we did not articulate that clearly. We did not claim “that substrates of the transporters make better inhibitors”. Rather, we wanted to state that “phenolic glucuronides that do not inhibit the uptake of prototypical substrates are not substrates of OATP.” We expected to use this characteristic as a fast screening tool to determine whether a glucuronide is not a substrate of a specific OATP. The latter will save our time to develop an analysis method to measure the glucuronide. This has been updated in the section 3.2.4.

Page 22, paragraph 1:

“The results showed that if a compound was not an inhibitor of prototypical OATP substrate, it was not a substrate of that OATP (Figure 3D1 to 3D3), but the reverse was not true. In contrast, substrates of OATP1B1 were able to significantly inhibited (*P<.005*) the uptake of glucuronides (Won-7-G, E2G and Lut-3’-G) that are good substrates of a specific OATP. Interestingly, aglycones were better inhibitors than their corresponding glucuronides, even though the corresponding glucuronides had higher LRE%.”

– The authors need to elaborate further in section 3.4 on the lack of physiological relevance for the 18/25 examples where there was no correlation between cellular uptake and RR% and the 7 which were not relevant with respect to relative expression of OATPs in human liver. Does the latter have something to do with Table S4? If so, that table should be referenced here.

We modified section 3.4 to further elaborate our statistical analysis methods to explain the physiological relevance as suggested by the reviewer.

The following is the revised changes along with page and paragraph information.

Page 23, paragraph 1:

“A total of 66 combinations with different OATP weightings were applied in Fisher exact test (Appendix 4 supplement table S1). OATP 1B1 is the most abundant OATP expressed in livers (Appendix 2 Supplement Table S1) [28] and also considered the most important among all OATPs.

OATP 1B3 is grouped with OATP1B1 in both localization and substrates, but the contribution of OATP 1B3 is considered to be always smaller than OATP 1B1. Therefore, combinations which assigned the highest weightings to OATP1B3 were considered not physiologically relevant and eliminated from the results. Of all the results from the Fisher exact test (Appendix 4 supplement Table S1), 22 combinations were eliminated based on this criterion. Of the remaining 44 combinations, 41 out of 44 results (93.2%) showed statistical significance. Taken together, these results demonstrated there were solid correlation between cellular uptake and LRE%. Thus, we showed that cellular uptake of glucuronides is the rate-limiting step in HER. In addition, this correlation (Figure 5) could be mathematically described by an E_max_ model (E_max_ 86%±17%) with an EC_50_ value of 42±33 nM.”

Reference

1. Fang, Y., et al., Study of Structure and Permeability Relationship of Flavonoids in Caco2 Cells. Nutrients, 2017. 9(12).

2. Tian, X.J., et al., Studies of intestinal permeability of 36 flavonoids using Caco-2 cell monolayer model. Int J Pharm, 2009. 367(1-2): p. 58-64.

3. Ahmed, I., et al., Caco-2 Cell Permeability of Flavonoids and Saponins from Gynostemma pentaphyllum: the Immortal Herb. ACS Omega, 2020. 5(34): p. 2156121569.

4. Singh, R., et al., Uridine diphosphate glucuronosyltransferase isoform-dependent regiospecificity of glucuronidation of flavonoids. J Agric Food Chem, 2011. 59(13): p. 7452-64.

5. Yuan, B., et al., Role of metabolism in the effects of genistein and its phase II conjugates on the growth of human breast cell lines. AAPS J, 2012. 14(2): p. 329-44.

6. Mohos, V., et al., Effects of Chrysin and Its Major Conjugated Metabolites Chrysin-7Sulfate and Chrysin-7-Glucuronide on Cytochrome P450 Enzymes and on OATP, P-gp, BCRP, and MRP2 Transporters. Drug Metab Dispos, 2020. 48(10): p. 1064-1073.

7. Griffiths, L.A. and G.E. Smith, Metabolism of apigenin and related compounds in the rat. Metabolite formation in vivo and by the intestinal microflora in vitro. Biochem J, 1972. 128(4): p. 901-11.

8. Chen, X., et al., Quantitation of the flavonoid wogonin and its major metabolite wogonin7 β-D-glucuronide in rat plasma by liquid chromatography-tandem mass spectrometry. J Chromatogr B Analyt Technol Biomed Life Sci, 2002. 775(2): p. 169-78.

9. Zhang, L., et al., Hepatic metabolism and disposition of baicalein via the coupling of conjugation enzymes and transporters-in vitro and in vivo evidences. AAPS J, 2011. 13(3): p. 378-89.

10. Hou, W.Y., et al., Age- and sex-related differences of organic anion-transporting polypeptide gene expression in livers of rats. Toxicol Appl Pharmacol, 2014. 280(2): p. 370-7.

11. Wang, L., et al., Metabolic Disposition of Luteolin Is Mediated by the Interplay of UDPGlucuronosyltransferases and Catechol-O-Methyltransferases in Rats. Drug Metab Dispos, 2017. 45(3): p. 306-315.

– Please describe the relevance of the blood stability test. This seems something included to satisfy a previous reviewer without any reference to the relevance of the finding to the authors' overall conclusion.

[Editors' note: further revisions were suggested prior to acceptance, as described below.]

Essential revisions:1) Please provide a reference for the statement in the last sentence of the introduction on line 135.

The statement is based on results shown in two papers. We used these references because in paper 1, it stated that raloxifene “is excreted almost exclusively as glucuronide conjugates via the feces of patients” 1[]. Since the metabolites of raloxifene are made of Ral-6-G and Ral-4’-G, with the latter represents 70% of its metabolites 2[], we made the statement that phenolic drugs are “significantly metabolized in the human intestine (up to ~70% or more) before entering the liver…”

The reason why the major metabolite of raloxifene Ral-4’-G is only formed in the human small intestine is because UGT isoforms UGT1A8 and UGT1A10 are responsible for its formation 34[, ], and both isoforms are predominantly expressed in the human enterocytes 5[]. Hence, small intestine was considered as being responsible for up to 70% of raloxifene metabolism in vivo.

Similar results showed that 78% of orally dosed ezetimibe could be found in feces of patients 6[]. Since intestine is the major metabolism organ of ezetimibe 7[], we believed that a large portion of that 78% (found in feces) was due to metabolism in the small intestine and subsequent excretion via HER. Our data in this paper also confirmed that ezetimibe was extensively metabolized in rat small intestine and underwent HER almost exclusively.

Genistein, a dietary flavonoid, is another compound extensively metabolized in the small intestine, and in humans only 10~30% of the dose was found in urine, suggesting 70-90% of the dose could be eliminated via feces in humans 8[].

2) Please define meaning of "LRE%" abbreviation upon its first use on line 140.

The content has been edited as showed below:

“The portal vein infusion uses a direct method to assess the recirculation efficiency of a phenolic glucuronide by the liver using liver recycle efficiency% (LRE%, defined as steady-state biliary secretion rate divided by the hepatic infusion rate).”

3) Please change recycle rate to LRE% on line 554.

We found an incorrect term on line 548, and changed the text from “RR%” to “LRE%”. However the term “excretion rate at the steady-state” on line 548 is not as clear as it should be, and therefore we replaced it with “steady-state glucuronide excretion rate”, which is the same term used in Equation 1.

References.

1. Teeter, J.S. and R.D. Meyerhoff, Environmental fate and chemistry of raloxifene hydrochloride. Environ Toxicol Chem, 2002. 21(4): p. 729-36.

2. Sun, D., et al., Characterization of raloxifene glucuronidation: potential role of UGT1A8 genotype on raloxifene metabolism in vivo. Cancer Prev Res (Phila), 2013. 6(7): p. 719-30.

3. Kemp, D.C., P.W. Fan, and J.C. Stevens, Characterization of raloxifene glucuronidation in vitro: contribution of intestinal metabolism to presystemic clearance. Drug Metab Dispos, 2002. 30(6): p. 694-700.

4. Kokawa, Y., et al., Effect of UDP-glucuronosyltransferase 1A8 polymorphism on raloxifene glucuronidation. Eur J Pharm Sci, 2013. 49(2): p. 199-205.

5. Strassburg, C.P., M.P. Manns, and R.H. Tukey, Expression of the UDP-glucuronosyltransferase 1A locus in human colon. Identification and characterization of the novel extrahepatic UGT1A8. J Biol Chem, 1998. 273(15): p. 8719-26.

6. Patrick, J.E., et al., Disposition of the selective cholesterol absorption inhibitor ezetimibe in healthy male subjects. Drug Metab Dispos, 2002. 30(4): p. 430-7.

7. Kosoglou, T., et al., Ezetimibe: a review of its metabolism, pharmacokinetics and drug interactions. Clin Pharmacokinet, 2005. 44(5): p. 467-94.

8. Yang, Z., et al., Bioavailability and pharmacokinetics of genistein: mechanistic studies on its ADME. Anticancer Agents Med Chem, 2012. 12(10): p. 1264-80.